Corrected: Author correction

# Structural analysis of human ARS2 as a platform for co-transcriptional RNA sorting

Wiebke Manuela Schulze[1], Frank Stein[2], Mandy Rettel[2], Max Nanao[1,3] & Stephen Cusack [1]

ARS2 is a highly conserved metazoan protein involved in numerous aspects of nuclear RNA metabolism. As a direct partner of the nuclear cap-binding complex (CBC), it mediates interactions with diverse RNA processing and transport machineries in a transcript-dependent manner. Here, we present the human ARS2 crystal structure, which exhibits similarities and metazoan-specific differences to the plant homologue SERRATE, most notably an additional RRM domain. We present biochemical, biophysical and cellular inter-actome data comparing wild type and mutant ARS2 that identify regions critical for inter-actions with FLASH (involved in histone mRNA biogenesis), NCBP3 (a putative cap-binding protein involved in mRNA export) and single-stranded RNA. We show that FLASH and NCBP3 have overlapping binding sites on ARS2 and that CBC–ARS2–NCBP3 form a ternary complex that is mutually exclusive with CBC–ARS–PHAX (involved in snRNA export). Our results support that mutually exclusive higher-order CBC–ARS2 complexes are critical in determining Pol II transcript fate.

[1] European Molecular Biology Laboratory, Grenoble Outstation, 71 Avenue des Martyrs, CS 90181, Grenoble Cedex 9 38042, France. [2] European Molecular Biology Laboratory, Heidelberg, Meyerhofstraße 1, Heidelberg 69117, Germany. [3]Present address: ESRF-The European Synchrotron, Avenue des Martyrs 71, CS40220, Grenoble Cedex 9 38043, France. Correspondence and requests for materials should be addressed to S.C. (email: cusack@embl.fr)

Distinct classes of RNA polymerase II (Pol II) transcripts undergo different co-transcriptional maturation pathways, and it is thought that the heterodimeric nuclear cap-binding complex (CBC), which binds to the 5′ cap structure of all nascent Pol II transcripts, plays a role in recruiting the appropriate processing machineries[1]. Recent studies further suggest that arsenite resistance protein 2 (ARS2) frequently serves as an adapter protein between CBC-capped RNA complexes and various RNA maturation complexes, including those for 3′ end processing, RNA transport and nuclear exosomal degradation[2–5].

Human ARS2 (or SRRT, Serrate RNA effector molecule homologue) is a predominantly nuclear protein of around 871–884 residues depending on the isoform. It is highly conserved in metazoans, but close homologues also exist in plants, where it is known as SERRATE[6], as well as in *Schizosaccharomyces pombe*[7,8]. It was first discovered as a protein-mediating arsenic resistance[9] and was subsequently shown to be essential for early mammalian development and cellular proliferation probably via a role in RNA metabolism[2,10]. Knockdown of ARS2 in mammalian cells results in defects in 3′ end processing of certain Pol II transcripts such as histone mRNAs and snRNAs, as well as in reduced degradation of short-lived promoter upstream transcripts (PROMPTs)[3,4,11,12]. It was shown that for correct histone mRNA processing, the direct interaction of ARS2 with Fas-associated death domain (FADD)-like IL-1β-converting enzyme (FLICE) associated with a huge protein (FLASH) is necessary[13–15]. Additionally, the interaction of ARS2 with DROSHA links the CBC–ARS2 complex to the miRNA-processing pathway[2,11,14], and this role of ARS2/SERRATE as part of the microprocessor complex is particularly important in plants[16]. Furthermore, in *Drosophila,* ARS2 interacts with Dicer-2 and is involved in siRNA biogenesis[17]. The important role that ARS2 plays in promoting correct processing of replication-dependent histone mRNA as well as in miRNA biogenesis likely explains its requirement in development and cell-cycle progression[2,14]. Higher-order complexes of CBC–ARS2 involving the zinc-finger protein ZC3H18 target RNAs for degradation by the nuclear exosome via either the PAXT (poly(A) tail exosome targeting) or NEXT (nuclear exosome targeting) complexes[3,12,18–20]. This role of ARS2 is conserved in *S. pombe*[7,8]. In the case of correctly processed transcripts, CBC–ARS2 is involved in the recruitment of transport factors, for instance, PHAX (phosphorylated adapter for RNA export) in the case of snRNAs[4,21] and ALYREF in the case of mRNAs[22,23]. Emerging from such findings is the concept that CBC–ARS2 forms the hub of a dynamic network of mutually exclusive higher-order complexes that determine transcript fate[5,20,24]. For instance, structural and biochemical studies have shown that CBC forms mutually exclusive complexes either with the transcriptional pausing factor NELF-E[25] or with ARS2, probably at respectively earlier or later stages of transcription[24]. CBC–ARS2 can interact with either PHAX or ZC3H18, promoting either nuclear export (e.g. of snRNA) or degradation via the NEXT/PAXT pathways[5]. Similarly, in the case of mRNAs, ALYREF or exosome cofactor MTR4 compete for binding to CBC–ARS2 to promote either nuclear export or degradation, respectively[23].

Despite its central role in nuclear RNA metabolism and its interactions with numerous partner proteins, as well as RNA, the structure of human ARS2 and how this relates to its multiple functions is poorly understood. A partial structure is available of *Arabidopsis thaliana* SERRATE[26]; however, plant proteins lack the RNA recognition motif (RRM) domain (also known as the ribonucleoprotein (RNP) domain) found in metazoans and *S. pombe*, as well as several other metazoan-specific elaborations. Based on the SERRATE structure, a first functional dissection of human ARS2 was made, resulting in tentative assignments of regions that

interact with CBC, DROSHA, FLASH, histone mRNA and miR-NAs[14]. The CBC binding site of ARS2 was reported to map to the unstructured C-terminal region, and this was confirmed by a crystal structure showing how the C-terminal 20 residues of ARS2 bind to human CBC[24]. Here, we describe the crystal structure of the folded core of human ARS2 corresponding to residues 147–270 and 408–763, which are separated by a large unstructured region. We use structure-based mutants to analyse how ARS2 interacts with RNA and FLASH and some of the same mutants to identify where partner proteins bind using affinity purification from cellular extracts combined with mass spectrometry. We find that the RRM domain is required to bind RNA, whereas a basic patch in the C-terminal leg of ARS2 mediates the interaction with FLASH. Additionally, we describe a novel CBC–ARS2 complex with nuclear cap-binding protein 3 (NCBP3), a putative additional cap-binding protein[27] and show that NCBP3 also binds to the C-terminal leg of ARS2 in a way that is mutually exclusive with the binding of PHAX to CBC–ARS2. These results provide the structural basis for further studies to clarify the detailed molecular mechanisms that make ARS2 a key adaptor protein in numerous co-transcriptional pathways.

## Results

**ARS2 crystal structure determination.** We set out to determine the crystal structure of human ARS2 and first worked with the construct ARS2$^{147–763}$, thus excluding the N- and C-terminal tails predicted to be unstructured. Since crystals of ARS2$^{147–763}$ only diffracted to around 20 Å resolution, and disorder predictions suggested the presence of an additional central, low-complexity region (Supplementary Fig. 1a), limited proteolysis was used to delineate the structured domains. Trypsination followed by size-exclusion chromatography (SEC) led to the identification of the stable core of ARS2 comprising two co-eluting fragments, each of which ran as a doublet (Supplementary Fig. 1b). In accordance with the disorder predictions, mass-spectrometric analysis showed that the two doublets comprise residues 147–285/290 and 393/405–763 (Supplementary Fig. 1b). Crystals (space group $P6_4$) of the core derived by trypsination diffracted to only 7 Å resolution probably because of the fragment heterogeneity. Better crystals (space group $P3_121$), diffracting to 3.7 Å resolution, were obtained by co-expression of two defined fragments denoted as ARS2$^{147–270+408–763}$ in *E. coli*. The structure of ARS2$^{147–270+408–763}$ was solved using selenomethionine-labelled protein and the single-isomorphous replacement with anomalous-scattering (SIRAS) method. The resultant map showed two molecules in the asymmetric unit and could be partially interpreted based on the known structure of the plant homologue SERRATE (PDB:3AX1 [10.2210/pdb3AX1/pdb]). Based on the incomplete model, the N-terminal helix was truncated at position 171 with the aim of producing better crystals. ARS2$^{171–270+408–763}$ crystallised in hexagonal space groups $P6_522$ or $P6_5$, diffracting to 3.2 and 3.5 Å resolution, respectively. Selenomethionine-labelled protein was used to solve the structure of the $P6_522$ form using the single-isomorphous replacement methods with anomalous scattering. This required merging data from several crystals to obtain a sufficiently strong anomalous signal (see Methods and Supplementary Fig. 2). A nearly complete model was built, making use of information from maps calculated in each different crystal form. Based on this model, additional constructs for crystallisation were designed in which disordered loops A (538–552) and B (568–598) (Fig. 1a, Supplementary Fig. 3) were truncated. Crystals of ARS2$^{147–270+408–763Δ568–598}$ (Δloop B) diffracted to 3.37 Å resolution (space group $C222_1$) and exhibited an ordered loop A. A composite model, including both loop A (from the ARS2$^{147–270+408–763Δ568–598}$ structure) and

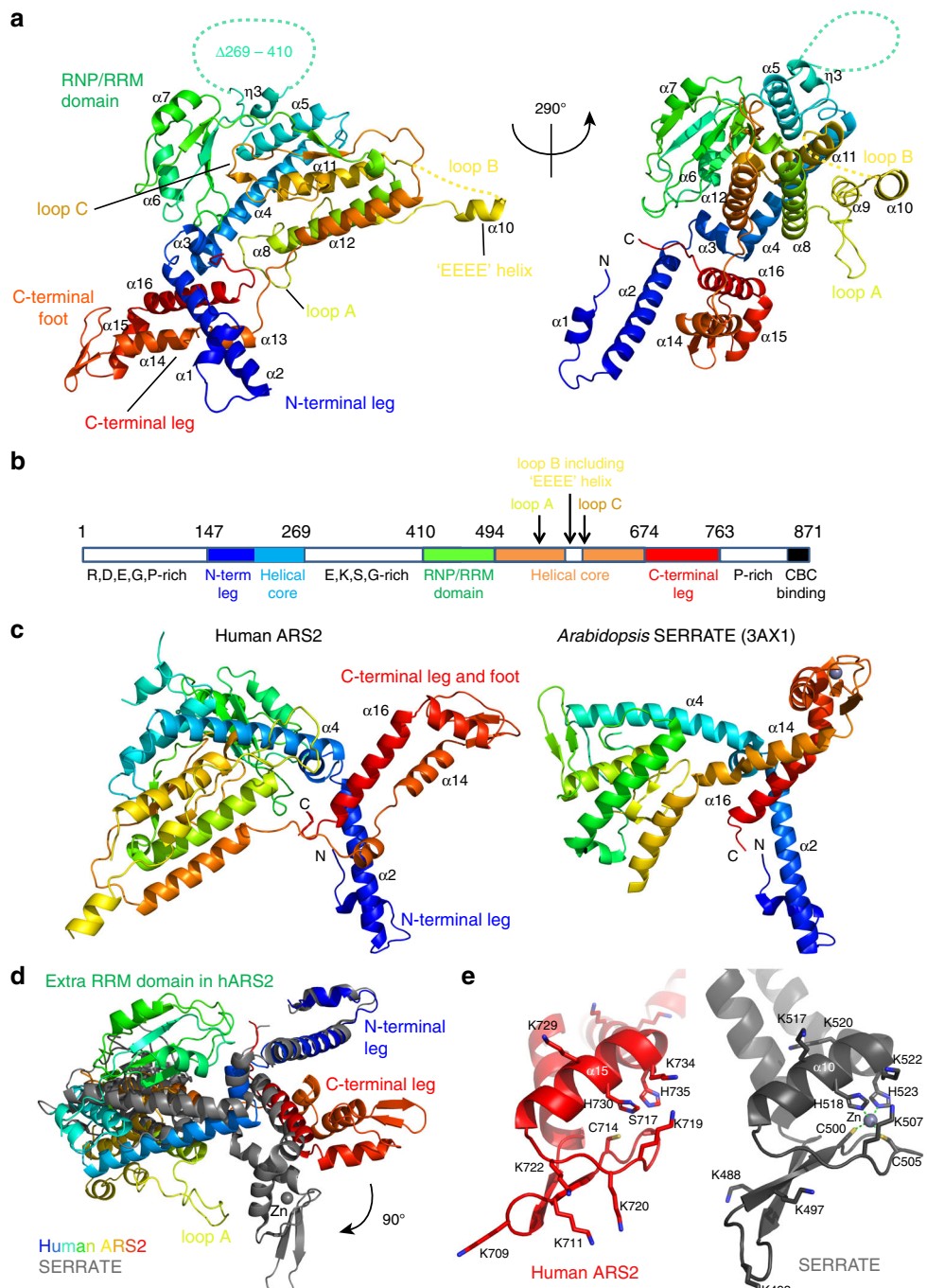

**Fig. 1** Crystal structure of human ARS2. **a** Composite ARS2$^{147-763}$ model showing loop A (residues 538–554) and 'EEEE' helix from loop B (residues 567–599). The structure is rainbow coloured from blue (N-terminus) to red (C-terminus). Dashed lines indicate unobserved or deleted regions. Annotated secondary-structure elements correspond to those on the structure-based alignment (Supplementary Fig. 3). **b** Schematic diagram showing the ordered regions (coloured) that are separated by unobserved and presumably, intrinsically disordered regions (white) (Supplementary Fig. 1b). **c** Comparison of human ARS2 (left) and *A. thaliana* SERRATE (right) crystal structures. Both structures are rainbow coloured as in **a**. The structures were aligned via the N-terminal helices regions since many of the other elements are differently orientated. A full structural alignment using DALI yields $Z = 14.4$, RMSD = 9.6 Å for 264 residues aligned with 25% identity. **d** Superposition of the human ARS2 (rainbow coloured) and SERRATE (grey) structures via their N-terminal regions, as in **c**. The 90° difference in the orientation of the C-terminal foot is highlighted. The grey sphere in the SERRATE structure is a zinc ion. **e** Differences between the C-terminal foot of metazoan ARS2 (left, red) and plant SERRATE (right, grey) shown in the same orientation. Key lysine residues are indicated, as well as the two cysteines (C500, C505) and two histidines (H518, H523) that co-ordinate the zinc ion (grey sphere) in SERRATE. In human ARS2, the equivalent of C505 is S717 and no zinc ion is bound

**Table 1 Crystallographic data collection and refinement statistics**

| Crystal | Human ARS2147–270+408–763 | Human ARS2171–270+408–763 | Human ARS2171–270+408–763 | Human ARS2 147–270+408–763 Δloop B: Δ568–598 E567–GSGSGS–E599 |
|---|---|---|---|---|
| PDB ID | 6F7P | 6F7J | 6F8D | 6F7S |
| *Diffraction data* | | | | |
| Beamline | ID30A1 | ID29 | ID29 | ID30A1 |
| Space group | $P3_121$ | $P6_522$ | $P6_5$ | $C222_1$ |
| Cell dimensions | $a = b = 136.64$ Å | $a = b = 90.92$ Å | $a = b = 105.21$ Å | $a = 85.51$ Å |
| | $c = 158.79$ Å | $c = 267.02$ Å | $c = 267.04$ Å | $b = 148.27$ Å |
| | $\alpha = \beta = 90°$ | $\alpha = \beta = 90°$ | $\alpha = \beta = 90°$ | $c = 235.66$ Å |
| | $\gamma = 120°$ | $\gamma = 120°$ | $\gamma = 120°$ | $\alpha = \beta = \gamma = 90°$ |
| Wavelength (Å) | 0.966 | 0.984 | 0.984 | 0.966 |
| Resolution range of data (last shell) (Å) | 50.0–3.7 (3.8–3.7) | 50.0–3.22 (3.3–3.22) | 50.0–3.54 (3.64–3.54) | 50.0–3.37 (3.46–3.37) |
| Completeness (last shell) (%) | 99.7 (98.2) | 97.0 (99.9) | 98.7 (99.4) | 94.3 (67.1) |
| R-sym (last shell) (%) | 8.1 (115.8) | 25.9 (167.5) | 10.1 (146.7) | 9.8 (67.4) |
| $I/\sigma I$ (last shell) | 12.11 (1.44) | 10.15 (1.42) | 8.62 (1.01) | 12.62 (1.97) |
| CC(1/2) (last shell) (%) | 100 (76.3) | 99.6 (54.4) | 99.8 (34.3) | 99.8 (53.7) |
| Redundancy (last shell) | 6.33 (6.16) | 10.13 (9.13) | 3.13 (3.20) | 4.12 (3.82) |
| *Refinement* | | | | |
| Reflections used in refinement: work (free) | 17,679 (979) | 10,435 (516) | 19,210 (879) | 19,276 (1017) |
| R-work (last shell) | 0.274 (0.432) | 0.289 (0.390) | 0.234 (0.405) | 0.229 (0.360) |
| R-free (last shell) | 0.309 (0.442) | 0.322 (0.455) | 0.263 (0.425) | 0.271 (0.389) |
| Number of non-hydrogen atoms | 7259 | 3445 | 6504 | 7225 |
| *Geometry and B-factors* | | | | |
| RMSD (bonds) (Å) | 0.007 | 0.007 | 0.008 | 0.008 |
| RMSD (angles) (°) | 1.014 | 0.937 | 1.195 | 1.159 |
| *Ramachandran* | | | | |
| Favoured (%) | 91.8 | 94.8 | 91.9 | 92.3 |
| Outliers (%) | 1.63 | 0.5 | 0.26 | 0.12 |
| Clash score | 0.96 | 1.45 | 1.07 | 2.14 |
| Molprobity score | 1.58 | 1.39 | 1.71 | 1.88 |
| Wilson B-factor (Å$^2$) | 150.7 | 69.6 | 123.2 | 96.4 |
| Average B-factor (Å$^2$) | 206.1 | 83.5 | 164.6 | 128.0 |

most of loop B (from the ARS2$^{147–270+408–763}$ structure) was artificially constructed and used for illustrative purposes since it is almost complete, only lacking residues 578–598 (Fig. 1). Data collection and refinement statistics are summarised in Table 1.

**Overall structure of the core of human ARS2.** The human ARS2 core has an unusual structure comprising a largely helical 'body' from which project N- and C-terminal 'legs' (Fig. 1). The body is made up of five roughly parallel α-helices (α4, α5, α8, α11 and α12), onto which is packed the metazoan-specific RRM domain. The latter has its helical surface solvent exposed, while the β-sheet surface is partially packed against the core (see below). Helices α1 and α2 form a protruding N-terminal leg and at the other extremity, helices α14–16 form a C-terminal leg that extends out in a different direction. The C-terminal leg ends in a 'foot'-like structure that features an exposed sole comprising a three-stranded β-sheet (Fig. 1a). The core helices are connected by long loops, some of which make extensive intramolecular interactions that hold the structure together (e.g. loop C between α11 and α12), while others form solvent-exposed, partially disordered loops (e.g. loop A and loop B between α8 and α11) (Fig. 1a, b). Between α5 and the RRM domain is a large insertion of ~140 residues (269–410), predicted to be unstructured due to its biased composition (E, K, S and G-rich) and which was deleted in the crystallised construct.

A database search reveals that only the homologue SERRATE from *Arabidopsis*[26] has a similar overall architecture to human ARS2 (Fig. 1c, d). However, human ARS2 differs from plant SERRATE in being overall more elaborate with many of the equivalent helices being orientated differently, as is apparent when the structures are superposed via helices α1–3 (Fig. 1c, d). The most significant difference is that SERRATE lacks the RRM domain that is present in metazoan ARS2 (Fig. 1d, Supplementary Fig. 3). Apart from this, the C-terminal leg is orientated quite differently with the foot rotated by ~90° (Fig. 1d, e). In plant SERRATE, C500, C505, H518 and H523 ligate, a zinc ion that structurally stabilises the foot, and these residues are also conserved in *S. pombe* ARS2. However, in metazoans, C505 is replaced by a serine (S717 in human ARS2), and no zinc is found in the structure (Fig. 1e). It should be borne in mind that the observed crystal structures of human ARS2 and plant SERRATE represent 95.6% and 86% of the crystallised constructs, respectively, but only 52% (457/871) and 42% (300/720) of the complete proteins. In particular, for human ARS2, the construct lacks the N- and C-terminal extensions and the large central region, all presumed to be unstructured (Fig. 1b, Supplementary Fig. 1a).

Calculation of the electrostatic potential shows that there are two distinct basic surface patches (Fig. 2a). The first is a positively charged cradle formed by the RRM β-sheet (including R425, R458, R459 and R463, as well as K418, R470 and K475) and the C-terminal extension of the RRM, notably residues R497, R501 and R504 (Fig. 2a, left). The second patch is on the sole of the C-terminal foot, which is very lysine rich (Fig. 2a, right).

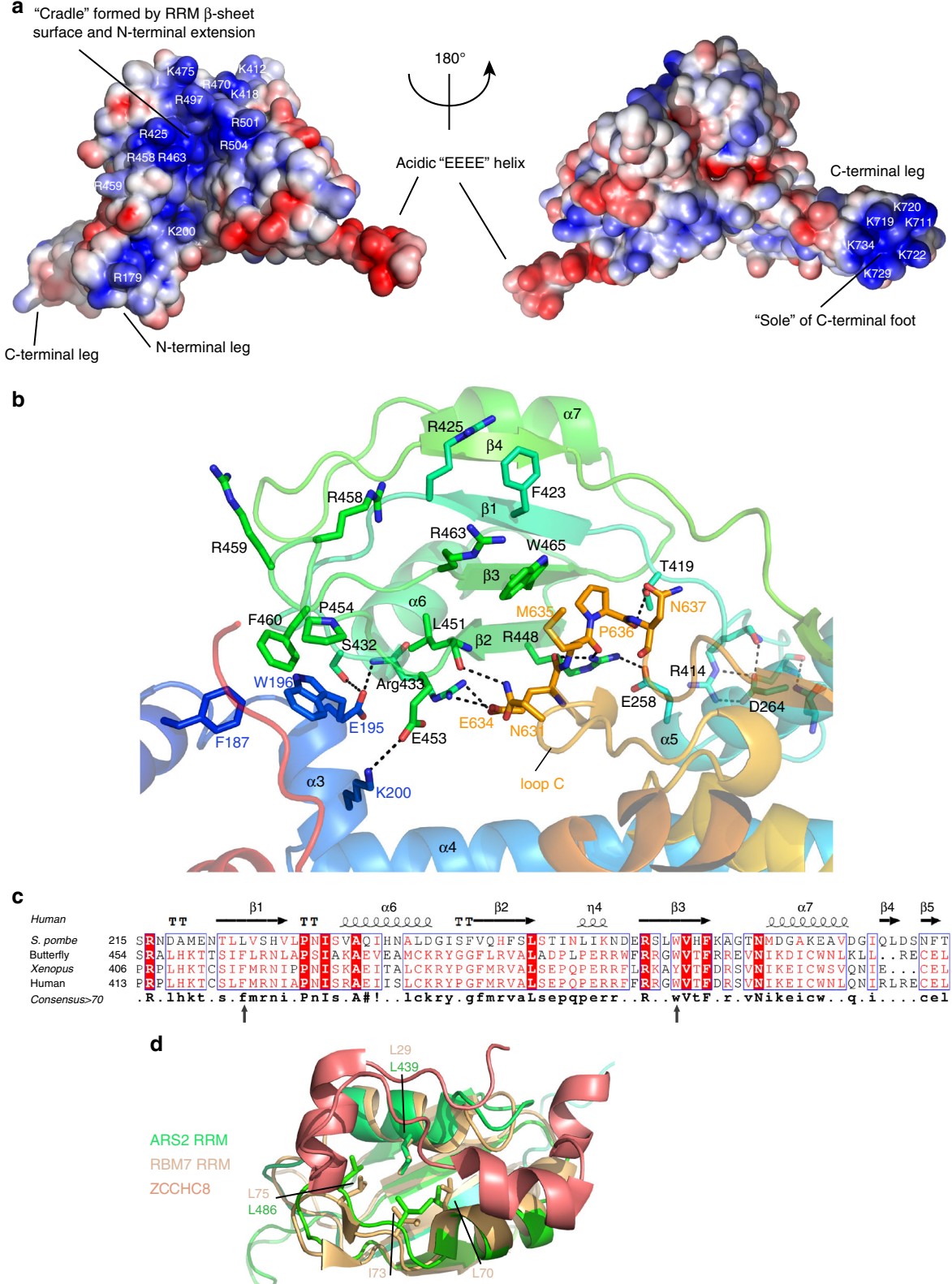

**Fig. 2** Structural features of the ARS2 RRM domain. **a** Electrostatic surface potential (red −ve, blue +ve) calculated in PyMOL for two orientations of human ARS2. Basic residues contributing to the two negatively charged patches are shown, as well as the positively charged 'EEEE' helix in loop B. **b** View of the RRM domain showing the exposed β-sheet residues R463 and W465 (RNP1) and F423 and R425 (RNP2) and interactions holding the RRM in place, notably via loop C residues. Putative hydrogen bonds are indicated with black dotted lines. **c** Sequence alignment of the *S. pombe*, butterfly, *Xenopus* and human RRM domain of ARS2 with superposed secondary structure. Conserved aromatic residues within the RNP motifs of metazoans are indicated with arrows. **d** Superposition of ARS2 RRM domain (green ribbons) with the RRM domain of RBM7 (yellow) bound to a ZCCHC8 peptide (salmon) (PDB:5LXY [10.2210/pdb5LXY/pdb]). Identical residues forming a hydrophobic surface that in the case of RBM7 binds the ZCCHC8 peptide are displayed as sticks

**RRM domain.** RRM domains are typically of around 80–90 residues and comprise a four-stranded antiparallel β-sheet upon which are packed two α-helices with a $\beta_1\alpha_1\beta_2\beta_3\alpha_2\beta_4$ topology. The exposed β-sheet surface frequently functions to bind short ssRNA motifs sequence specifically and to facilitate stacking of RNA bases, the central regions of β1 (RNP2 motif) and β3 (RNP1 motif) that often carry exposed aromatic residues[28]. However, there are many cases known where RRM domains exclusively mediate protein–protein interactions with such contacts being made by either the β-sheet or α-helical faces[29]. The ARS2 RRM domain (residues 411–500) is highly conserved in all metazoans (Fig. 2b). It has one conserved aromatic and basic residue in each of the RNP motifs; for hARS2, these are F423/R425 in β1 and R463/W465 in β3 (exception K/Y in β3 in *Xenopus*). The domain is also present in *S. pombe*, but with a highly divergent sequence (Fig. 2c). The RRM domain is firmly held in place by hydrophobic and polar interactions with four regions of the core of ARS2 (187–200, 251–268, 631–641 and 759–762), burying 2456 Å² of the solvent-accessible surface (PISA server) (Fig. 2b). Of particular note is loop C (residues 631–641) which is six residues longer in metazoans (and *S. pombe*) compared to plants (Supplementary Fig. 3), enabling it to pack partially against the RRM β-sheet. M635 and P636 are in contact with one face of W465 and E634, making a salt-bridge with R433. Other van der Waals interactions include P454 and F460 (in the β2–β3 loop) with F187 and W196 and I255 with L426. R448 from β2 makes a salt bridge with E258, as well as hydrogen bonds to the main-chain carbonyls of loop-C residues 634–635. The α-helical face of the RRM domain is solvent exposed with four hydrophobic residues (L439, L481, I484 and L486) identically positioned as in the RRM domain of the RBM7 subunit of the nuclear exosome targeting (NEXT) complex (L29, L70, I73, L75 and PDB:5LXY [https://doi.org/10.2210/pdb5LXY/pdb]) (Fig. 2d). Since RBM7 uses this surface for protein–protein interactions with a peptide from another NEXT component, ZCCHC8[19], this observation suggests that ARS2 might also interact with partner proteins via the α-helical face of the RRM domain.

**Metazoan-specific loops.** Apart from loop C, two other metazoan-specific loops are of interest. In mammals, loop A (538-ASEPGT₅₄₃PPLPTSLPS-552) is highly conserved and proline-rich. In *Xenopus* and *S. pombe*, the equivalent loop is shorter and in insects it is longer, but in neither case is it proline-rich. In most crystal forms, the loop is disordered presumably due to flexibility. However, in the crystal form in which loop B (see below) was deleted, loop A is well structured in one monomer, likely due to crystal-packing constraints (Fig. 1a). Interestingly, T543 is a major phosphorylation site of human ARS2 (Discussion).

Loop B (residues 566–598) is around 30-residues long in metazoans, half the length in *S. pombe* and absent in plants (Supplementary Fig. 3). In metazoans, the first part of the loop is highly acidic (566-EEVSAEEEELLG-577), whereas the second half of the loop is proline-rich, particularly in mammals. In all crystal structures of hARS2 (except that with loop B deleted), only the first part of the loop is observed with 570-AEEEELL forming helix α10 ('EEEE helix'). Remarkably, all crystal forms containing loop B, irrespective of space-group, exhibit the same intermolecular crystal contact, in which the acidic 'EEEE helix' and the preceding short β-strand pack into the positively charged cradle formed by the RRM β-sheet and the C-terminal extension of the RRM (Supplementary Fig. 4a). To accommodate the same intermolecular contact in different crystal-packing environments, some angular flexibility of the loop B structure is required (Supplementary Fig. 4b).

**The ARS2 RRM domain preferentially binds single-stranded RNA.** We used a fluorescence polarisation assay to study the binding of ARS2 to various FAM-labelled model RNAs as well as ARS2 mutants in which either of the two basic patches, the RRM

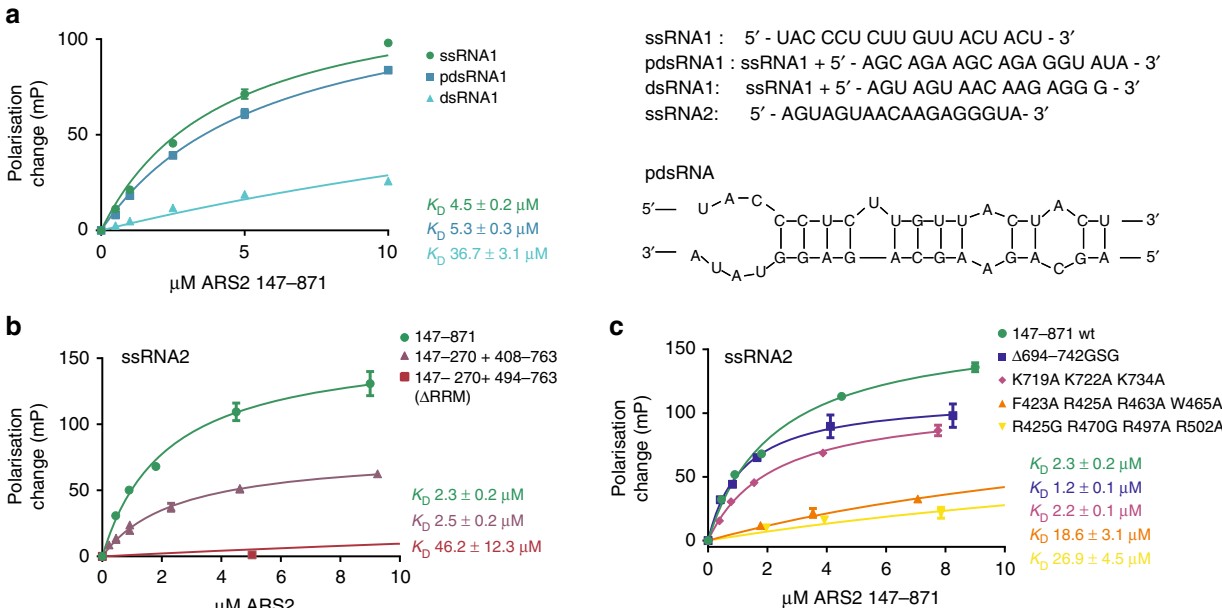

**Fig. 3** The RRM domain of ARS2 binds single-stranded RNA. FAM-labelled RNA and fluorescence polarisation measurements were used to determine the affinity of various RNAs to hARS2. See also Supplementary Fig. 5. **a** Recombinant ARS2^147–871 was titrated to FAM-labelled single-stranded (ss), partially double-stranded (pds) and double-stranded (ds) RNA and the polarisation measured for each concentration. **b** ARS2^147–871, ARS2^147–270+408–763 and ARS2^147–270+494–763 were titrated to FAM-labelled ssRNA and the fluorescence polarisation measured. **c** Site-specific mutants of basic residues in the RRM domain, or mutants and deletion in the C-terminal foot of ARS2^147–871, as indicated, were used to further localise the RNA-binding site. The $K_D$ was derived by fitting the data to a single-site binding model. Error bars show SD of three experiments

**Table 2 Summary of dissociation constants measured by ITC**

| Interaction analysed | $K_D$ (µM) | Number of ITC experiments |
|---|---|---|
| ARS2 147–871 + FARB [HS] | 5.1 ± 1.1 | 2 |
| ARS2 147–871 + FLASH 921–946 [HS] | 5.5 ± 0.7 | 2 |
| ARS2 147–871 + scrambled FARB [HS] | n.d. | 2 |
| ARS2 147–871 K719A K722A K734A + FARB [HS] | n.d. | 2 |
| ARS2 147–871 Δ692–743 GSG + FARB [HS] | n.d. | 2 |
| ARS2 171–270 + 408–763 + FARB [HS] | 3.8 ± 0.3 | 3 |
| ARS2 147–270 + 494–763 + FARB | 0.5 ± 0.14 | 2 |
| ARS2 147–871 + NCBP3 1–282 [HS] | Interaction visible, but $K_D$ n.d. | 2 |
| ARS2 147–871 K719A K722A K734A + NCBP3 1–282 [HS] | n.d. | 2 |
| ARS2 147–871 Δ692–743 GSG + NCBP3 1–282 [HS] | n.d. | 2 |
| NCBP3 1–282 + m$^7$GTP | 8.3 ± 3.0 | 3 |
| NCBP3 + m$^7$GTP | 4.6 ± 1.0 | 3 |
| NCBP3 + m$^7$GTP [HS] | 5.1 ± 0.6 | 3 |
| NCBP3/2xCBP80 + m$^7$GTP [HS] | 7.9 ± 0.9 | 2 |
| NCBP3 1–282 + GTP | n.d. | 2 |
| CBP20 + m$^7$GTP [HS] | 84 ± 4.2 | 2 |
| CBP20/2xCBP80 + m$^7$GTP [HS] | 0.37 ± 0.04 | 2 |

[HS] high salt. ITC experiments were performed in buffer containing 250 mM NaCl
$K_D$ values and standard deviation represent the average from at least two independent experiments as indicated

β-sheet surface and the C-terminal foot (Fig. 2a) were perturbed. We determined a $K_D$ of 2–5 µM for either UC or AG-rich single-stranded (ss) RNAs binding to ARS2$^{147-871}$ (Fig. 3). A similar value was found for partially double-stranded (pds) RNA, but was much lower for dsRNA (Fig. 3a). The crystallised core, ARS2$^{147-270+408-763}$, which lacks the central disordered region, retained identical ssRNA binding, whereas for ARS2$^{147-270+494-763}$, missing the RRM domain as well, no significant RNA binding was detected with concentrations up to 50 µM ARS2 protein (Fig. 3b, Supplementary Fig. 5a). Furthermore, point mutations of the conserved and exposed RRM β-sheet basic and aromatic residues (ARS2$^{147-871 \, F423A \, R425A \, R463A \, W465A}$) and of other basic residues in the extended basic patch (ARS2$^{147-871 \, R425G \, R470G \, R497A \, R502A}$) (Fig. 2a) also resulted in a significantly reduced ssRNA affinity, whereas mutations in the basic patch of the C-terminal foot (ARS2$^{147-871 \, K719A \, K722A \, K734A}$), as well as its deletion (ARS2$^{147-871 \, Δ694-742GSG}$), retained ssRNA binding (Fig. 3c). We therefore conclude that the RRM β-sheet surface and nearby basic residues, rather than the C-terminal foot, are critical for ARS2 binding to ssRNA.

**Mapping the binding site of FLASH on ARS2.** It was shown previously that a central 13-amino acid peptide of FLASH (931-DELEEGEIRSDSE-943), called FARB (FLASH–ARS2-binding region), is sufficient for the interaction with ARS2[13]. However, the exact ARS2 residues involved in this interaction have not been identified, although the RRM and C-terminal leg have been implicated[14]. Using isothermal titration calorimetry (ITC), we determined an affinity of 5 µM of FARB, as well as of the longer construct FLASH$^{921-946}$, for ARS2$^{147-871}$, at 250 mM NaCl (the $K_D$s derived by ITC are summarised in Table 2). In contrast, following the work of Kiriyama et al.[13], a scrambled peptide, 'EGEIESELSEDDR', containing the same residues randomly rearranged and thus showing the same acidity as the original FARB peptide resulted in no measurable interaction with ARS2$^{147-871}$ (Fig. 4a). The binding site was further narrowed down to ARS2$^{147-270+494-763}$ (Fig. 4a), which lacks the unstructured regions as well as the RRM domain. To identify the FLASH interaction site on ARS2 by pull down, FLASH$^{903-943}$ that contains FARB, was cloned into the GST-vector pETM30 and expressed and purified from E. coli. After immobilising GST–FLASH$^{903-943}$ on glutathione sepharose and incubating with different structure based ARS2 mutants, analysis of the

elutions showed that ARS2$^{Δ694-742GSG}$ (lacking the C-terminal leg) and ARS2$^{K719A \, K722A \, K734A}$ do not bind GST–FLASH$^{903-943}$ (Fig. 4b, Supplementary Fig. 6). ITC experiments with these two constructs confirmed that both abolished the interaction with FARB (Table 2, Supplementary Fig. 6a). Thus, the C-terminal leg, in particular, three lysines of the basic patch on the foot, mediate the ARS2 binding to FLASH.

**Identification of ARS2 interaction partners.** To gain further insight into which ARS2 domains or loops interact with which partner proteins, different ARS2 constructs were expressed as EGFP-tagged fusion proteins in HEK 293T/17 cells and affinity purification/mass spectrometry (AP-MS) used to identify interaction partners. To achieve this, EGFP followed by a GSGGGS linker and the ARS2 coding sequence were cloned into pcDNA3.1. The EGFP tag enabled direct visualisation of the transfection efficiency and cellular localisation of the ARS2 variants. All constructs localised in the nucleus similar to the wild type (Supplementary Fig. 7). After immunoprecipitation of the overexpressed EGFP–ARS2, the eluted fractions were tandem mass tag (TMT) labelled and analysed via mass spectrometry. Wild type ARS2 showed enrichment for numerous proteins from complexes involved in many aspects of RNA metabolism (Fig. 5a, Supplementary Dataset 1). Components of the RNA degradation machineries (NEXT (RBM7/ZCCHC8/hMTR4)), PAXT (PABPN1/ZFC3H1/hMTR4) and exosomal components (EXOSC4, EXOSC8, EXOSC9 and EXOSC10) were identified, in accordance with previous studies[3,4,23,30]. Additionally, splicing factors (THRAP3, TRA2A/B and snRNP proteins), DDX helicases, 3′ end processing factors (CPSF1/2/3/3L/6/7) as well as hnRNP components, subunits of the TRanscription and EXport (TREX) complex (THOC1, 2, 3, 5 and 6), and ALY, hnRNPC and PHAX that are involved in the RNA export, were also identified. As expected, CBC components CBP80 (NCBP1) and CBP20 (NCBP2) were highly enriched, but also NCBP3 (previously C17orf85 or ELG), an mRNA-binding protein[31,32] recently found to associate with CBP80 and to be important for mRNA export[27].

To confirm that the method could be used to identify regions of ARS2 that mediate specific protein–protein interactions, we first introduced mutations into ARS2 C-terminus (ARS2$^{1-871 \, R854A \, R859A \, F871A}$ and ARS2$^{1-845}$), which were shown previously in vitro to abolish the interaction with CBC[24]. Analysis of the co-precipitating proteins showed that both mutants failed to pull

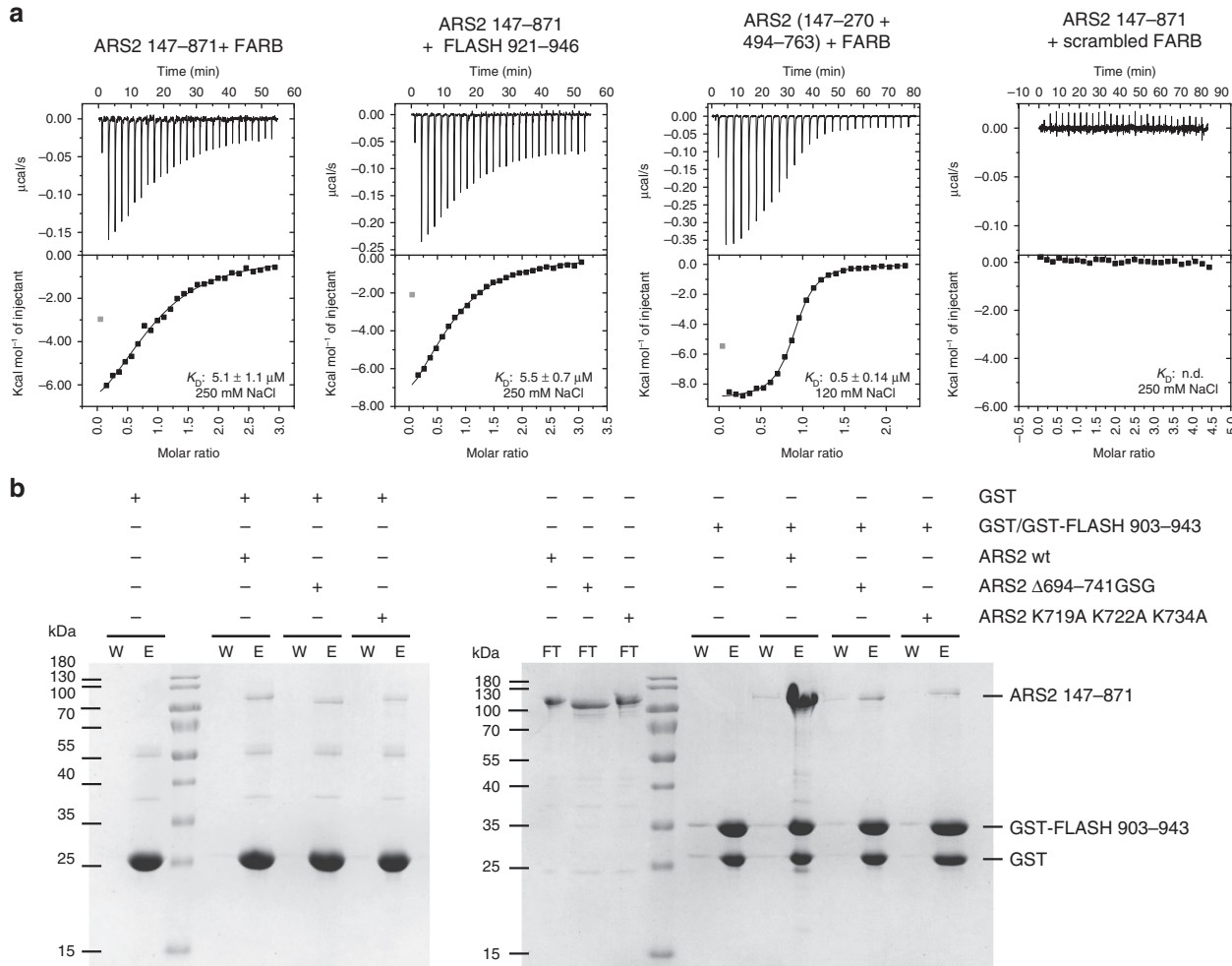

**Fig. 4** The FARB peptide of FLASH binds to the ARS2 C-terminal leg. **a** Isothermal calorimetry data for different ARS2 constructs and FLASH peptides. The upper graphs show the raw data and the bottom graphs show the ligand concentration dependence of the heat released upon binding after normalisation. $K_D$ values represent the average from at least two independent experiments (Table 2). **b** GST pull down of GST–FLASH[903-943] and different ARS2 constructs. GST–FLASH[903-943] or GST as control was immobilised on glutathione resin followed by incubation with different ARS2 constructs. The flow-though (FT), last wash (W) and elution (E) fractions were analysed by Coomassie-stained SDS-PAGE

down CBC as expected, but also no longer PHAX (Fig. 5b), which is a direct interaction partner of CBC.

We then used the C-terminal leg deletion or basic patch lysine mutants, ARS2[Δ694–742GSG] or ARS2[K719A K722A K734A], respectively, and observed that they co-precipitated fewer proteins compared to wild type ARS2 (Fig. 5c). Unlike wild type ARS2, neither construct showed enrichment for PHAX (suggesting that it may have a direct interaction with ARS2 as well as CBC), nor components of the NEXT and PAXT complexes, nor of the mRNA export machinery (ALY, THO components) and NCBP3, whereas their association with CBP80 and CBP20 was not affected. These results indicate that the C-terminal leg of ARS2 and, in particular its basic patch, mediate many protein–protein interactions and is possibly important for the mutually exclusive recruitment of distinct RNA-processing machineries. Note that although we show above that FLASH interacts with this region of ARS2, it was never identified in our pull downs or those of others[3,4], likely due to its cell-cycle dependence.

**NCBP3 can bind to ARS2 and CBP80 simultaneously.** We chose to characterise one ARS2-interacting protein that we

identified, NCBP3, in more detail. In a recent study, NCBP3, which contains a putative N-terminal RRM domain, was found to bind the cap-analogue m7GTP as well interact directly with CBP80 and ARS2[27]. It was proposed that NCBP3 could substitute for CBP20/NCBP2 and form an alternative cap-binding complex with CBP80/NCBP1 that was important for mRNA export[27]. To determine how NCBP3 interacts with ARS2 and CBC, we used recombinant full-length NCBP3 (residues 1–620) and the N-terminal region NCBP3[1–282], which contains the RRM domain. Both full-length NCBP3 and NCBP3[1–282] bound the cap analogue m7GTP with a $K_D$ of 5–8 μM (Supplementary Fig. 8a, Table 2). This is comparable, but less than the reported $K_D$ of 16 μM determined by microscale thermophoresis experiments for NCBP3[1–282] at a salt concentration of 225 mM[27] whereas we used 120 mM NaCl. Our results confirm that the C-terminal residues 283–620 are not required for cap-binding. In the case of CBC, CBP20 alone binds relatively weakly to the cap but the binding is considerably enhanced when CBP20 is bound to CBP80. We confirmed this by ITC at 250 mM salt concentration for m7GTP binding to CBP20 or CBC and found a more than 200-fold enhancement in affinity upon CBP20–CBP80 heterodimer formation (Supplementary Fig. 8b, Table 2). In contrast, no

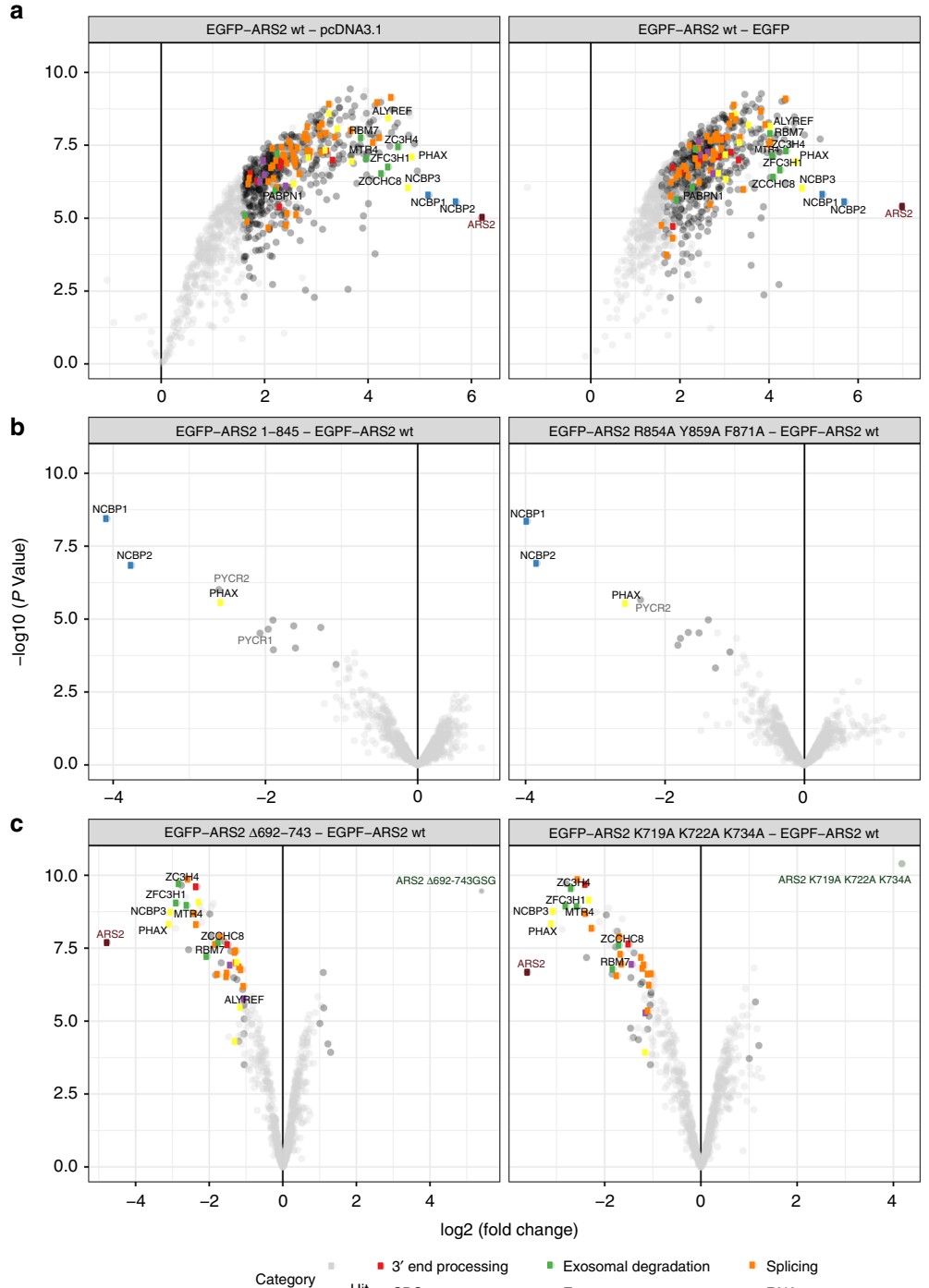

**Fig. 5** Identification of cellular interacting partners of ARS2. HEK 293T/17 cells were transfected with different EGFP-tagged ARS2 constructs or EGFP or pcDNA3.1 as control. After EGFP pull down, the associated proteins were analysed by LC–MS/MS. Volcano plots show the enrichment of EGFP–ARS2 wt associated proteins against the enrichment of the proteins for: **a** pcDNA3.1 (left) and EGFP (right); and the enrichment of proteins associated with ARS2 mutants against ARS2 wt: **b** EGFP–ARS2$^{1-845}$ (left) and EGFP–ARS2$^{R854A\ R859A\ F871A}$ (right). **c** EGFP–ARS2$^{\Delta692-743GSG}$ (left) and EGFP–ARS2$^{K719A\ K722A\ K734A}$ (right). Three independent affinity purifications were performed for each bait, except for the ARS2 wt in **b** for which only one pull down was performed. Bait proteins (dark red for wild type and dark green for mutant), CBC (blue), exosomal proteins (purple) and proteins involved in 3′ end processing (red), nuclear exosomal degradation (green), splicing (orange) and RNA transport (yellow) are highlighted. A full list of identified proteins is given in Supplementary Data 1

enhancement of the relatively modest cap-binding affinity of NCBP3 was observed upon addition of CBP80 under the same conditions (Supplementary Fig. 8c, Table 2).

To further characterise its interactions, NCBP3 was immobilised on m⁷GTP sepharose and incubated with various ARS2 constructs. Elution from the resin shows that ARS2$^{147-871}$ (Fig. 6a), as well as various loop-deletion constructs used for crystallisation (ARS2$^{171-270+408-763}$, ARS2$^{147-270+408-763\Delta loopB}$ and ARS2$^{171-270+408-763\Delta loopA\Delta loopB}$, Supplementary Fig. 9) are all retained by the NCBP3 bound to the resin. When CBP80 was

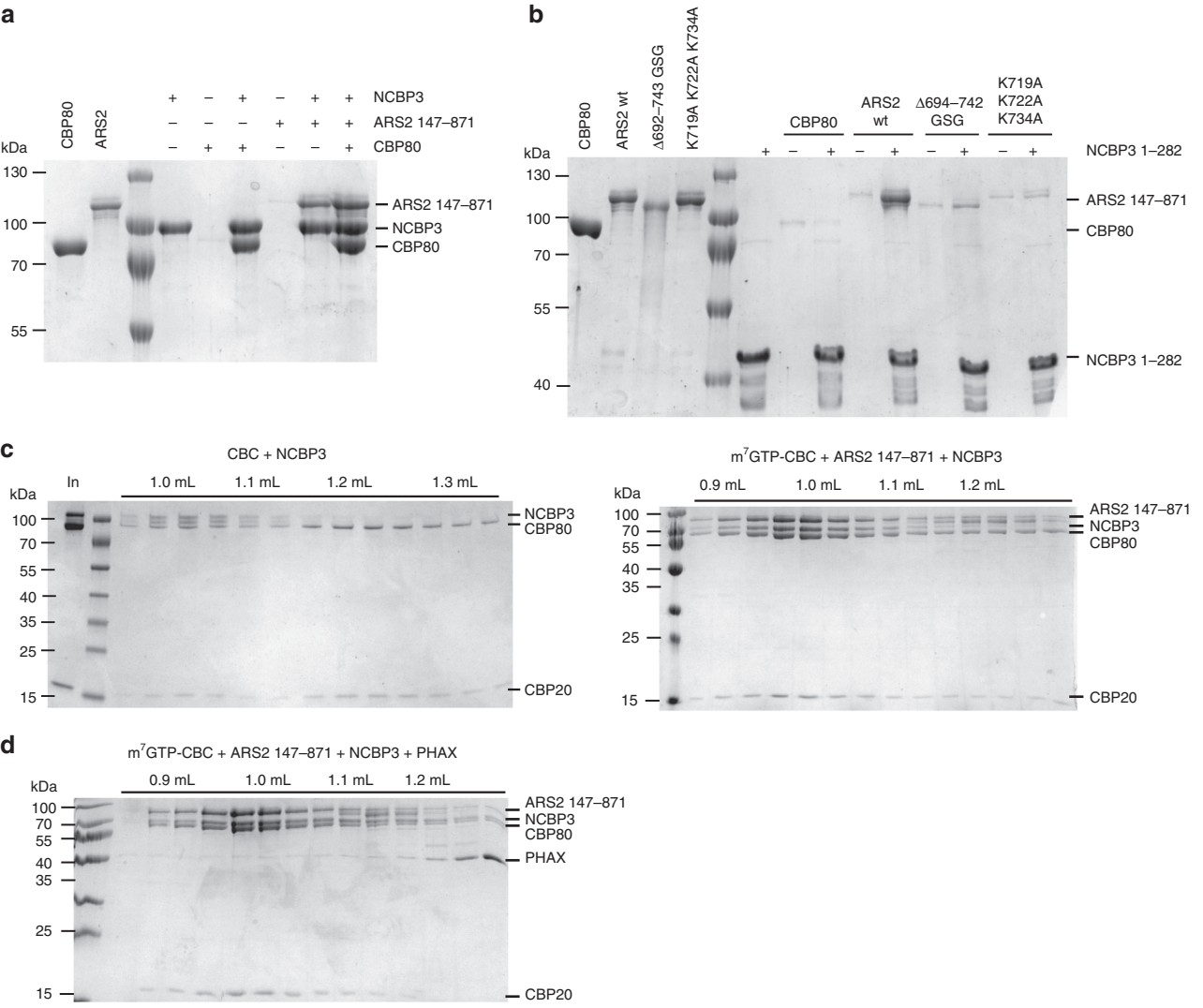

**Fig. 6** NCBP3 forms a CBC–ARS2–NCBP3 complex. **a**, **b** m$^7$GTP pull-down of full-length NCBP3 (**a**) or NCBP3$^{1-282}$ (**b**) with ARS2 and/or CBP80. Purified recombinant NCBP3 constructs were immobilised on m$^7$GTP sepharose before incubation with CBP80 and/or wild type or mutant ARS2$^{147-871}$. Eluted fractions were analysed by Commassie-stained SDS-PAGE. **c** Coomassie-stained SDS-PAGE of CBC–NCBP3 and CBC–ARS2–NCBP3 complexes analysed by gel filtration. The purified recombinant proteins were mixed, subjected to gel filtration and the individual protein-containing fractions were analysed by SDS-PAGE. **d** Coomassie-stained SDS-PAGE of CBC–ARS2–PHAX plus NCBP3 analysed by gel filtration. CBC–ARS2–PHAX was mixed with molar excess of NCBP3, subjected to gel filtration and the individual fractions were analysed by SDS-PAGE

included in addition to ARS2$^{147-871}$, CBP80 was also retained (Fig. 6a). The interaction with CBP80 was only observed for full-length NCBP3 and not for the N-terminal construct containing only the putative cap-binding domain (Fig. 6b). However, NCBP3$^{1-282}$ was sufficient to bind ARS2$^{147-871}$ (Fig. 6b). Furthermore, NCBP3$^{1-620}$ and NCBP3$^{1-282}$ co-eluted with GST-tagged ARS2$^{147-871}$ in a GST pull-down experiment in the absence of the cap analogue (Supplementary Fig. 10a), indicating that the interaction between NCBP3 and ARS2 is cap-independent and mediated via the N-terminal 282 amino acids of NCBP3. In addition to the dimeric complexes, co-elution of all three proteins NCBP3, CBP80 and ARS2 was also observed (Fig. 6a). This trimeric complex as well as the NCBP3–CBP80 and NCBP3–ARS2 complex could be reconstituted in the absence of m$^7$GTP, as indicated by a shift in elution volume on SEC (Supplementary Fig. 10c).

The observation within the EGFP–ARS2 pull-down experiments that ARS2$^{\Delta694-742GSG}$ and ARS2$^{K719A\ K722A\ K734A}$ are less enriched in NCBP3 than wild-type ARS2 (Fig. 5c), suggests that

the C-terminal leg of ARS2 might mediate the ARS2–NCBP3 interaction. To confirm this, ARS2$^{147-871}$ or mutants were tested for their ability to bind NCBP3 in vitro via m$^7$GTP pull-down. As shown in Fig. 6b, ARS2$^{K719A\ K722A\ K734A}$ and ARS2$^{\Delta694-742GSG}$ did not bind to the m$^7$GTP-bound NCBP3. This result was also confirmed by ITC, in which no binding of ARS2$^{K719A\ K722A\ K734A}$ to NCBP3$^{1-282}$ could be detected (Supplementary Fig. 10b). These results together with the AP–MS of the different ARS2 constructs indicate that the basic patch on the C-terminal leg of ARS2 mediates its interaction with the N-terminal region of NCBP3. Thus, NCBP3 and FLASH have overlapping binding sites on ARS2.

**CBC–ARS2–NCBP3 form a ternary complex.** Next, we questioned whether the binding of NCBP3 to CBP80 and ARS2 is compatible with the formation of the complete CBC heterodimer, which shows a 50-fold higher affinity for the m$^7$GTP-cap analogue compared to NCBP3 (NCBP3 $K_D$ ~5 µM; CBC $K_D$ ~0.1 µM,

Supplementary Fig. 10b)[24]. Upon addition of NCBP3 to CBC and subjecting the mixture to SEC, we found that NCBP3 co-eluted with CBC (Fig. 6c, left). Carrying out the same experiment in the presence of m$^7$GTP and with ARS2$^{147-871}$ in addition resulted in the co-elution of all four proteins (Fig. 6c, right). It is well-established that CBC–ARS2–PHAX complex form a stable ternary complex[4,24]. To test whether PHAX is compatible with NCBP3 binding to CBC–ARS2 we performed further SEC experiments. These show that the CBC–ARS2–NCBP3–PHAX complex cannot be formed and that NCBP3 is able to displace PHAX from a preformed CBC–ARS2–PHAX complex (Fig. 6d). Taken together these results show that CBC–ARS2–NCBP3 and CBC–ARS2–PHAX form mutually exclusive ternary complexes.

## Discussion

ARS2 is a direct interaction partner of CBC and together they form a platform for the assembly of co-transcriptional complexes that determine the fate of diverse RNA Pol II transcripts. However, detailed structural information on human ARS2 is lacking with only a preliminary functional dissection of the domain structure having been made[14]. In this work, we determined the crystal structure of the human ARS2 and used the structural information to shed light on how ARS2 interacts with partner proteins. Our structural analysis shows that ARS2 is made up of two regions, residues 147–270 and 408–763 that co-fold into a structured core with an intervening glutamate-rich region of unknown function that is likely unstructured. In addition, there are N- and C-terminal unstructured extensions to the core, the former containing the nuclear localisation sequence (NLS)[14] and the extreme C-terminus binding to CBC[24]. The core has several structural features in common with the plant protein SERRATE, notably the N- and C-terminal legs, but the latter is orientated quite differently and the positively charged C-terminal foot in metazoans is not stabilised by a zinc ion, unlike in plants. However, there are several other metazoan-specific elaborations, most importantly the RRM domain, which is held in place by an extended loop C, as well as surface loops A and B that are respectively proline-rich or contain a solvent exposed acidic helix ('EEEE' helix α10). The exact function of loops A and B remain to be determined. However, interestingly T543 within loop A is reported to be a major phosphorylation site of hARS2 (https://www.phosphosite.org/proteinAction.action?id=2872&showAllSites=true)[33] possibly by a proline-directed kinase such as casein kinase 2 (CK2)[34]. This may be a means of regulating ARS2 activity. Concerning loop B, in most ARS2 crystal forms the negatively charged 'EEEE' helix binds onto the positively charged RRM β-sheet surface of a neighbouring molecule (Supplementary Fig. 4). Repetition of this inter-molecular contact generates a three-dimensional network. However, it appears sterically impossible for one molecule to intra-molecularly 'auto-inhibit' its own RNA binding surface in this way, although it could potentially happen inter-molecularly if ARS2 is at a sufficiently high concentration.

Overall, our biochemical studies demonstrate that the crystallised core ARS2$^{147/171-270+408-763}$ has the same activity as ARS2$^{147-871}$ with respect to several of the biological functions of ARS2 including interactions with ssRNA (Fig. 3b), FLASH (Fig. 4a, Table 2) and NCBP3 (Fig. 6, Supplementary Fig. 9), and further deletions or mutations enable the interacting regions to be more precisely defined. Thus we show that the positively charged β-sheet surface of the RRM can bind various ssRNAs with micromolar affinity whereas the basic C-terminal foot is not involved in such binding (Fig. 3). This result is somewhat at variance with previous results showing that the positively charged

C-terminal leg (designated ZnF[14]), and possibly the N-terminal leg (designated DUF3546[14]), are important for binding more elaborate RNAs such as histone mRNA or miRNA and that the β-sheet surface of the RRM was not involved in such binding[14]. We note in passing that in this previous work, which was based on homology modelling of ARS2, the authors misidentified the RNP1 motif of the RRM domain, predicting that F468 (which is actually buried) and R470 corresponded to the exposed residues rather than R463 and W465 in reality[14]. Clearly more studies, particularly structural, are required to define the exact mode of binding of particular RNAs to ARS2 and whether there is any RNA sequence or structural specificity.

Our pull downs with EGFP–ARS2, and similar work by others, show that ARS2 interacts with a large number of partner proteins involved in multiple aspects of nuclear RNA metabolism. Several of these, such as CBC, FLASH, DROSHA, ZC3H18, MTR4, ALYREF and NCBP3 make demonstrably direct interactions with ARS2[3,4,13,14,23,27,30]. The structural basis for these interactions is largely unknown, except for the case of CBC, where a crystal structure showing that the C-terminal 20 residues of ARS2 bind at the CBP20/CBP80 interface has recently been determined[24]. The ARS2 structure shows that the α-helical surface of the RRM is solvent exposed and thus potentially involved in mediating protein–protein interactions with partner proteins. In this respect it is interesting that a PDB search reveals that the RRM domain structurally most similar to that of ARS2 is in NEXT subunit RBM7 ($Q = 0.57$, $Z = 7.1$, RMSD 1.55 Å). The RBM7 structure was determined in complex with a peptide from another NEXT component ZCCHC8, which binds to hydrophobic residues on the RBM7 RRM α-helical surface. Intriguingly, the equivalent ARS2 surface has a very similar hydrophobic patch (Fig. 2d), but it remains to be seen which, if any, partner proteins bind here.

Several lines of evidence implicate the ARS2 C-terminal leg in a number of different protein–protein interactions. We find that the acidic FARB peptide of FLASH binds to basic residues on the C-terminal foot (Fig. 4b), consistent with previous results[14]. However, these previous results also implicated the β-sheet surface of the ARS2 RRM in FARB binding. It is intriguing that in most of our structures, the acidic 'EEEE helix' (566-EEV-SAEEEELLG-577) within loop B, which has a certain sequence similarity to FARB (931-DELEEGEIRSDSE-943) can make an inter-molecular interaction with the β-sheet surface of the RRM from another ARS2 molecule, suggesting that FARB might bind in the same place. Nevertheless, despite extensive co-crystallisation trials, including with the loop B/'EEEE' helix deletion mutant, we have never observed any electron density corresponding to FARB bound to ARS2 in a crystal, and all our biochemical evidence suggests that only the C-terminal foot is involved. Our results confirm previous studies[27] identifying NCBP3 as a direct interactor of ARS2 and furthermore show that this interaction implicates the RRM-containing N-terminal region of NCBP3 and the C-terminal leg and foot of ARS2 (Fig. 6b), suggesting that NCBP3 and FLASH may bind in a mutually exclusive fashion to ARS2.

NCBP3 (C17orf85/ELG) was recently proposed to form a novel cap-binding complex with CBP80 that is important for mRNA export[27]. Although we confirm that NCBP3 binds the cap-analogue m$^7$GTP, the affinity is 50 fold weaker than for CBC and is not enhanced when bound to CBP80, unlike for CBP20. Furthermore, we show that in vitro it is possible to reconstitute CBC–NCBP3 and CBC–ARS2–NCBP3 complexes consistent with the earlier identification of NCBP3 (as ELG) as a CBC-dependent component of mRNPs[31]. Indeed both CBP80 and CBP20 as well as ARS2 and TREX and EJC components were identified as associated proteins of NCBP3 in previous AP–MS

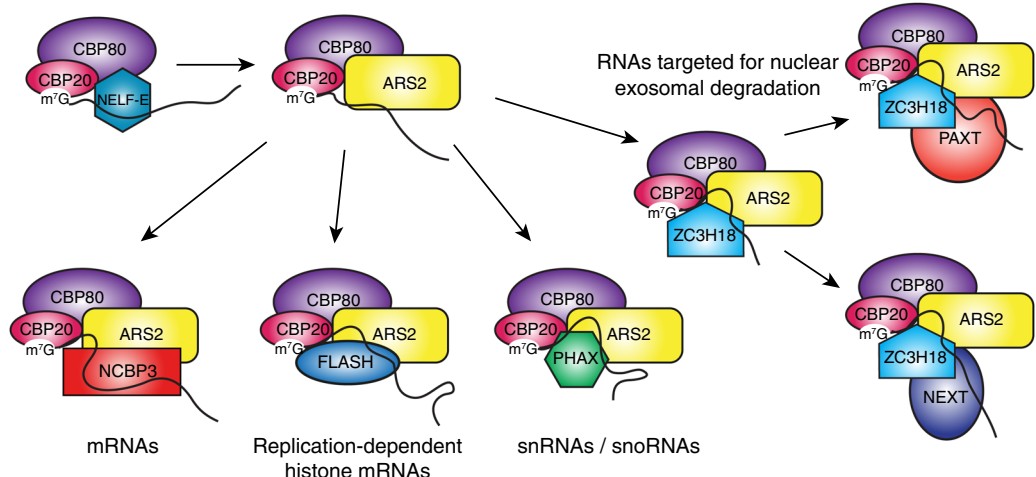

**Fig. 7** Mutually exclusive CBC–ARS2 complexes. Schematic diagram showing model of mutually exclusive CBC and CBC–ARS2 complexes identified in this work or by others (see text). Early in transcription CBC is proposed to bind to NELF-E which is subsequently replaced by ARS2[24]. Then, depending on the type of RNA and the corresponding 3′ end processing and export pathways, alternative and mutually exclusive CBC–ARS2 complexes assemble. Other complexes target the RNA for exosomal degradation

experiments[27], and NCBP3 (as C17orf85) was independently found to associate with ALYREF[23]. These results support NCBP3's function, in combination with CBC, to promote mRNA splicing and/or export[27,31]. Our finding that the CBC–ARS2–NCBP3 complex is not able to bind PHAX, in agreement with the demonstration that snRNA is not associated with NCBP3[27], suggests that one role of NCBP3 might be to exclude inappropriate transport factors from mRNA. Further studies are required to clarify the function of the weak cap-binding activity of NCBP3 including whether it is enhanced by factors other than CBP80, although it appears that NCBP3 can substitute for CBP20/NCBP2 in certain conditions[27].

The notion that mutually exclusive CBC or CBC–ARS2 complexes co-transcriptional assembly on RNAs requiring different biogenesis pathways is now becoming widely accepted (Fig. 7). Earlier we showed that the binding of negative elongation factor subunit NELF-E and ARS2 to CBC is mutually exclusive[24], suggesting that CBC–NELF-E might precede CBC–ARS2 complex formation on the same transcript. Additionally, it was shown that the CBC–ARS2–PHAX complex, important for snRNA export, is incompatible with the CBC–ARS2–ZC3H18 complex that targets RNA for nuclear endosomal degradation[5]. Whether the transcript is then degraded via either the PAXT or NEXT pathway depends on the exclusive binding of CBC–ARS2–ZC3H18 to ZFC3H1/PABPN1/MTR4 or ZCCHC8/RBM7/MTR4, respectively[20]. Recently, it was also shown that there is a dynamic competition between binding of MTR4 or ALYREF to ARS2 promoting either mRNA degradation or export[23]. The mutually exclusive interaction of NCBP3 or PHAX with CBC–ARS2, shown in this work, also seems to be a factor in the sorting of mRNA or snRNA into different export pathways. On the other hand, histone mRNAs are directed into their specific maturation pathway by forming an exclusive complex with CBC–ARS2–FLASH. Our work suggests that the overlapping binding sites of different partner proteins on, for instance, the C-terminal leg of ARS2, is likely a critical aspect in defining these mutually exclusive complexes. The alternative suggestion that ARS2 undergoes significant conformational changes in response to different RNA/partner protein binding[14] seems less likely given the limited flexibility, apart from surface loops, so far observed for ARS2 (Supplementary Fig. 4). However, structural analysis of larger complexes together with the corresponding transcript will be needed to resolve this question.

In conclusion, our structural and biochemical analysis of human ARS2 is an important step forward in elucidating the dynamic protein-protein and protein–RNA interaction network that underlies the central role of CBC–ARS2 as a platform for contributing to Pol II transcript fate determination.

## Methods

**Protein expression and purification**. Residues 147–871, 147–270, 171–270 of human ARS2 isoform-e (871 residues, NP_001122326, also known as isoform-4: Q9BXP5-4), hPHAX (NM_032177.3) residues 1–394 and hNCBP3 (NP_001107590.1) residues 1–282, were cloned into pETM11 (EMBL). hARS2 (408–763, 494–763) was cloned for co-expression into pDUET and pET15b (Novagen), FLASH (NP_036247.1) (903–943) was cloned into pETM30 (EMBL) and full-length hNCBP3 (1–620) was cloned into pFASTBac HTb (Invitrogen). CBP20 and CBP80ΔNLS were expressed and purified separately or together (CBC) as described previously[24,35]. Specific mutations were introduced into the ARS2 plasmids using site-directed mutagenesis. Primers used in this work are listed in Supplementary Table 1.

All constructs were expressed in *E. coli* Rosetta 2 (Novagen) over night at 18 °C, except for full-length NCBP3, which was expressed in High Five insect cells (Invitrogen). Cells were harvested and lysed by sonication in Lysis buffer containing 50 mM HEPES pH 7.8, 300 mM NaCl, 10% (v/v) glycerol, 5 mM β-mercaptoethanol. Lysates were clarified by centrifugation (60,000 × g, 1 h, 10 °C) and applied to an equilibrated Ni-Sepharose column (GE Healthcare). While dialysing into 20 mM HEPES pH 7.8, 120 mM NaCl, 10% (v/v) glycerol, 5 mM β-mercaptoethanol, the N-terminal 6-histidine-tag was cleaved with his-tagged Tobacco Etch Virus (TEV) protease overnight (protein to protease ratio 1:75). The cleaved protein was further purified first on Ni-Sepharose to remove His-TEV and then using anion exchange column (HiTrap Q, GE Healthcare) followed if necessary by a Heparin column (HiTrap Heparin, GE Healthcare) and size exclusion chromatography (Superdex 200 10/300 GL, GE Healthcare).

For selenomethionine incorporation into ARS2 constructs, bacteria were grown in M9 medium supplemented with 1 mM MgSO₄, 0.1 mM CaCl₂, 0.4 g L⁻¹ glucose and 2 mg L⁻¹ thiamine sulphate. At an OD₆₀₀ₙₘ of 0.8, the temperature was lowered to 18 °C, 0.1 g L⁻¹ lysine, 0.1 g L⁻¹ phenylalanine, 0.1 g L⁻¹ threonine, 0.05 g L⁻¹ isoleucine, 0.05 g L⁻¹ valine and 0.05 g L⁻¹ selenomethionine were added and after 30 min incubation protein expression was induced with 0.3 mM isopropyl β-ᴅ-1-thiogalactopyranoside (IPTG).

FLASH (NP_036247.1) peptides (residues 931-943 (FARB), 921-946) were purchased from GeneScript.

Protein complexes were purified, after mixing the individual proteins, by size exclusion chromatography (Superdex 200 10/300 GL or Superdex 200 Increase 3.2/300, GE Healthcare) in 120 mM NaCl, 1 mM tris(2-carboxyethyl) phosphine (TCEP), 20 mM HEPES pH 7.8.

**Limited proteolysis followed by MS**. ARS2$^{147–763}$ was digested 2 h at room temperature with trypsin (enzyme to protein ratio 1:500) (Promega). After addition of cOmplete EDTA free protease inhibitor cocktail (Roche) the protein was subjected to SEC (Superdex 200 10/300 GL, GE Healthcare; 20 mM HEPES, 120 mM NaCl pH 7.8) and the protein containing fractions were analysed by Coomassie-

stained SDS-PAGE. Fragment bands were cut out and diced into 1 mm cubes prior to in-gel hydrolysis. Dehydration of gel pieces by acetonitrile enabled buffer exchange prior and after the reduction and alkylation of cysteine-containing proteins, which was carried out using the following conditions: dithiothreitol (56 °C, 30 min, 10 mM) and iodacetamide (room temperature, in the dark, 20 min, 55 mM), in 100 mM ammonium bicarbonate. After transferring the dehydrated gel pieces to glass tubes (300 μL), 20–30 μL 3 M HCl were added. After placing the tubes inside 2 mL tubes containing 700 μL water, the tubes were closed and the samples microwaved for 10 min at 900 W. Afterwards the supernatant was directly desalted on Oasis HLB Elution Plate (Waters) according to the manufacturer's instructions. Samples were eluted in 50 μL, dried in a speed vacuum centrifuge and reconstituted in 10 μL buffer (96:4 water: acetonitrile, 0.1% formic acid) and analysed by LC–MS/MS[36].

Peptides were separated using the nanoACQUITY UPLC system (Waters) fitted with a trapping column (nanoACQUITY Symmetry C18, 5 μm, 180 μm × 20 mm) and an analytical column (nanoACQUITY BEH C18, 1.7 μm, 75 μm × 200 mm). The outlet of the analytical column was coupled directly to a Linear Trap Quadrupole (LTQ) Orbitrap Velos Pro (Thermo Fisher Scientific) using the Proxeon nanospray source. Solvent A was 0.1% formic acid in water and solvent B was acetonitrile supplemented with 0.1% formic acid. The sample was loaded with a constant flow of solvent A at 5 μL min$^{-1}$ onto the trapping column. Peptides were eluted via the analytical column a constant flow of 0.3 μL min$^{-1}$. The percentage of solvent B increased in a linear fashion up to 40%. Peptide transfer was enabled by a Pico-Tip Emitter 360 μm OD × 20 μm ID; 10 μm tip (New Objective), on which a spray voltage of 2.2 kV was applied. The temperature of the capillary was set at 300 °C. Full scan mass spectra (MS$^1$) with mass range 300–1700 $m/z$ were acquired in profile mode in Fourier transform (FT) with a resolution of 30,000. The filling time was set at maximum of 500 ms with a limitation of 10$^6$ ions. The most intense ions (up to 15) from the MS$^1$ were selected for a fragment screen (MS$^2$) in the LTQ. A normalised collision energy of 40% was used, and the fragmentation was performed after accumulation of 3 × 10$^4$ ions or after filling time of 100 ms for each precursor ion. MS$^2$ data was acquired in centroid mode, only multiply charged (2$^+$, 3$^+$ and 4$^+$) precursor ions were selected. The dynamic exclusion list was restricted to 500 entries with maximum retention period of 30 s and relative mass window of 10 ppm. Mass accuracy was improved using a lock mass correction of a background ion ($m/z$ 445.12003). For filtering the data and creating.mgf files MaxQuant (version 1.0.13.13) was used. The files were used to search in MASCOT version 2.2.03 (Matrix Science) against the *Homo sapiens* Uniprot database with a list of common contaminants appended. The following parameters were set for the search: Carbamidomethyl (C) (fixed modification) and Oxidation (M) (variable modification), the MS$^1$ scan allowed a mass error tolerance of 20 ppm and for the MS$^2$ scan 0.5 Da, and no enzyme was specified. Peptide ladders of increasing length, either starting or ending at the same residue, were used to determine the termini.

**Isothermal titration calorimetry**. For isothermal titration calorimetry (ITC) experiments a MicroCal ITC200 system (Malvern) was used. To exclude buffer differences, all proteins were dialysed against 20 mM HEPES, 120 mM, 2 mM TCEP pH 7.8 overnight at 8 °C. Experiments were performed at 25 °C with a stirring speed of 800 rpm. At an interval of 120–180 s, 1.5 μL of protein A was injected into the cell containing 205 μL of protein B. For the binding between NCBP3 and ARS2 as well as ITC experiments to determine the affinity to m$^7$GTP, proteins were dialysed in buffer containing 250 mM NaCl. For the interaction between NCBP3 and ARS2, injections of 2.5 μL were used. Data were analysed using the MicroCal ITC Origin software and the binding curves were fitted to a single-site binding equation.

**Fluorescence polarisation measurements**. Fluorescence anisotropy binding assays were performed with 5′ FAM-labelled and non-labelled RNA obtained from IBA: 5′-FAM-AGUAGUAACAAGAGGGUA 3′; 5′-FAM-AGUAGUAACAA-GAGGGUA 3′; 5′-AGCAGAAGCAGAGGU 3′; 5′-AGUAGUAACAAGAGGG 3′. 37.5 nM RNA were titrated with increasing concentrations of proteins in 115 mM NaCl, 2 mM TCEP, 7.5% (v/v) glycerol, 1 mM MgCl$_2$, 20 mM HEPES pH 7.8. Fluorescence polarisation was measured at 25 °C with a microplate reader (CLARIOstar BMG LABTECH) using an excitation wavelength of 495 nm and emission wavelength of 515 nm. For analysis, the polarisation value for the RNA alone was subtracted from the measurements and the data fitted to a single-site binding model with GraphPad Prism.

**GST pull down**. Lysate containing GST, GST-tagged protein or purified protein was incubated with glutathione sepharose (GE Healthcare) for 1 h at 10 °C. After washing the in vitro expressed and purified potential binding partner were added and incubated rotating for 30 min at 10 °C followed by extensive washing and elution of the proteins bound to the resin with SDS loading dye.

**EGFP AP–MS**. For AP–MS EGFP-tagged constructs, including a GSGGGS linker, were cloned into pcDNA3.1 and expressed in 293T/17 cells (ATCC). Forty-eight hours post transfection cells were harvested and lysed on ice for 30 min in HEK Lysis Buffer (20 mM HEPES pH 7.4, 120 mM NaCl, 0.1% (v/v) Triton X-100)

supplemented with cOmplete protease inhibitor cocktail (Roche) and 1 μg mL$^{-1}$ RNase H (Thermo Fisher Scientific). After centrifugation (16,000 × $g$, 20 min, 4 °C) the supernatant was incubated for 1 h at 4 °C with pre-equilibrated Sepharose A beads (GE Healthcare) before incubation with 25 μL GFP-Trap (ChromoTek) beads at 4 °C for 2 h. After washing the beads three times with 500 μL HEK Lysis Buffer (400 × $g$, 2 min, 10 °C) proteins were eluted using 50 μL 0.1 M glycine pH 2.3 followed by a second elution with 30 μl. The eluted protein was neutralised with 0.1 M Tris-HCl pH 8.5.

**TMT sample preparation**. Samples were treated with dithiothreitol (56 °C, 30 min, 10 mM) to reduce disulphide bridges in cysteine containing proteins. Subsequently, reduced cysteines were alkylated with iodacetamide (room temperature, in the dark, 30 min, 10 mM). Prior LC–MS/MS, the samples were prepared using the SP3 protocol[37]. For digestion, trypsin (sequencing grade, Promega) was added in an enzyme to protein ratio 1:50, and incubated overnight at 37 °C. Resulting peptides were labelled with TMT10plex (DOI: 10.1021/ac500140s) Isobaric Label Reagent (Thermo Fisher Scientific) according the manufacturer's instructions. Sample clean-up was carried out on an OASIS HLB μElution Plate (Waters) following manufacturer's instructions. An Agilent 1200 Infinity high-performance liquid chromatography system, equipped with a Gemini C18 column (3 μm, 100 × 3.0 mm, 110 Å, Phenomenex), was used to perform offline high pH reverse phase fractionation.

**LC–MS/MS**. The UltiMate 3000 RSLC nano LC system (Dionex) fitted with a trapping cartridge (μ-Precolumn C18 PepMap 100, 5 μm, 300 μm i.d. x 5 mm, 100 Å) and an analytical column (Acclaim PepMap 100 75 μm × 50 cm C18, 3 μm, 100 Å) was used for peptide separation. Solvent A was 0.1% formic acid in water and solvent B was 0.1% formic acid in acetonitrile. Trapping was carried out with a constant flow of solvent A at 30 μL min$^{-1}$ onto the trapping column for 6 min. Subsequently, peptides were eluted with a constant flow of 0.3 μL min$^{-1}$ via the analytical column. Percentage of solvent B increased in a linear fashion from 2 to 4% in 4 min, from 4 to 8% in 2 min, then 8 to 28% for a further 96 min, and finally from 28 to 40% in another 10 min. The outlet of the analytical column was coupled directly to a QExactive Plus (Thermo Fisher Scientific) mass spectrometer using the ProXeon nanoflow source in positive ion mode.

The peptides were introduced into the QExactive Plus via a Pico-Tip Emitter 360 μm OD × 20 μm ID; 10 μm tip (New Objective) and an applied spray voltage of 2.3 kV. The capillary temperature was set at 320 °C. MS$^1$ spectra were acquired with mass range 350–1400 $m/z$ in profile mode in FT with resolution of 70,000. The filling time was set at maximum of 100 ms with a limitation of 3 × 10$^6$ ions. Data-dependent acquisition (DDA) was performed with the resolution of the Orbitrap set to 35,000, with a fill time of 120 ms and a limitation of 2 × 10$^5$ ions. A normalised collision energy of 32 was applied. A loop count of 10 with count 1 was used and a minimum automatic gain control (AGC) trigger of 2e$^2$ was set. Dynamic exclusion time of 30 s was used. The peptide match algorithm was set to 'preferred' and charge exclusion 'unassigned', charge states 1, 5–8 were excluded. MS$^2$ data was acquired in profile mode (https://doi.org/10.1016/j.ymben.2018.03.006).

**MS data analysis**. Acquired data was processed by IsobarQuant[38] and Mascot (v2.2.07) and searched against a Uniprot *Homo sapiens* proteome database (UP000005640) containing common contaminants, reversed sequences and construct sequences. The data was searched with the following modifications: Carbamidomethyl (C) and TMT10 (K) (fixed modification), Acetyl (N-term), Oxidation (M) and TMT10 (N-term) (variable modifications).

For MS$^1$ spectra a mass error tolerance of 10 ppm and for MS$^2$ spectra of 0.02 Da was set. Trypsin was selected as protease with an allowance of maximum two missed cleavages and a minimum peptide length of seven amino acids. At least two unique peptides were required for a protein identification. The false discovery rate on peptide and protein level was set to 0.01.

The raw output data of IsobarQuant was processed using the R programming language (ISBN 3-900051-07-0). Only proteins which were identified with at least two unique peptide matches were kept for downstream analysis. Subsequently, potential batch effects were removed using the LIMMA package[39] and data was normalised using the VSN package (variance stabilisation[40]). In order to identify proteins that were significantly differentially expressed, LIMMA was used again. A protein was called a hit with a false discovery rate smaller than 1% and a fold change of at least 50%. Furthermore, hits were filtered and only called hits if they were enriched three-fold with a false discovery rate smaller than 1% comparing EGFF-ARS2 wild type against the EGFP and pcDNA3.1 control and additionally showed a false discovery rate bigger than 1% in the comparison of EGFP against pcDNA3.1.

**Crystallisation**. Crystals of variants of the core of ARS2 were obtained at 4 °C in 2 μL hanging drops with a 1:1 ratio of protein solution (6 mg mL$^{-1}$ in 20 mM HEPES, 300 mM NaCl, 2 mM tris(2-carboxyethyl)phosphine pH 7.8) to crystallisation solution. For ARS2$^{147–270+408–763}$ and ARS2$^{147–270}$ $^{+408–763Δ566–598GSGSGS}$, the best crystals grew with a crystallisation solution containing 0.2 M potassium citrate tribasic monohydrate, 20% (w/v) PEG 3350, pH

8.3. For ARS2$^{171-270+408-763}$, the crystallisation solution contained 1.2 M sodium dihydrogen phosphate, 0.8 M dipotassium hydrogen phosphate, 0.1 M CAPS pH 10.5 and 0.2 M lithium sulphate. For ARS2$^{147-270+408-763\Delta539-554GSA}$ the crystallisation solution contained 0.1 M Bis–Tris propane pH 6.5 0.2 M lithium sulphate and 22% (w/v) PEG 3350. For ARS2$^{147-270+408-763}$ the crystallisation solution contained 0.2 M lithium sulphate and 20% (w/v) PEG 3550. Selenomethionine-substituted crystals of ARS2$^{147-270+408-763}$ and ARS2$^{171-270+408-763}$ grew under the same conditions as wild type proteins. In several crystallisation screens, FLASH$^{921-946}$ or FLASH$^{931-943}$ (FARB) were added, but the peptide was never observed in any structure.

**Structure determination.** For data collection, crystals were flash-frozen in well solution supplemented with 20% (v/v) glycerol and diffraction measured on European Synchrotron Radiation Facility (ESRF) beamlines, MASSIF-1 (ID30A1)[41] or ID29. Data were integrated and scaled using the XDS suite[42] and further analysed using the CCP4i suite[43].

For phasing of the $P6_522$ ARS2$^{171-270+408-763}$ structure, four selenomethionine data sets were selected using a genetic algorithm[44] that maximised the anomalous signal while retaining acceptable merging $R$-values. These data sets were merged in XSCALE and submitted to the CRANK2 pipeline[45] in combination with the native data set for substructure determination and phasing by SIRAS. The resulting map was used for automatic model building with BUCCANEER followed by iterative rounds of manual building and refinement with REFMAC5[46]. The well-defined selenomethionine positions were used as sequence markers (Supplementary Fig. 2).

Subsequent ARS2 structures were solved by molecular replacement with PHASER[47] using the ARS2$^{171-270+408-763}$ structure. Refinement was performed with REFMAC5[46] with local non-crystallographic symmetry restraints and map-sharpening. Structure figures were drawn with PyMOL[48].

**Data availability.** Structure factors and co-ordinates have been deposited in the wwPDB with codes as follows:
6F7P: ARS2$^{147-270+408-763}$ structure ($P3_121$ form), 6F7J: ARS2$^{171-270+408-763}$ structure ($P6_522$ form), 6F8D: ARS2$^{171-270+408-763}$ structure ($P6_5$ form), 6F7S: ARS2$^{147-270+408-763\,\Delta loop\,B}$ structure ($C222_1$ form). The mass spectrometry proteomics data have been deposited to the ProteomeXchange Consortium via the PRIDE partner repository with the dataset identifier PXD009035. Other data are available from the corresponding author upon reasonable request.

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

## Acknowledgements

We thank the staff of the ESRF-EMBL Joint Structural Biology Group, in particular Matthew Bowler, for access to and help on ESRF beamlines. We thank the EMBL Grenoble Eukaryotic Expression Facility (EEF) and high-throughput crystallisation facility (HTX), the EMBL Heidelberg Proteomics Core Facility and Luca Signor from the Mass Spectrometry Platform at IBS. This work used the platforms of the Grenoble Instruct Center (ISBG: UMS 3518 CNRS-CEA-UJF-EMBL) within the Grenoble Partnership for Structural Biology (PSB), with support from FRISBI (ANR-10-INSB-05-02) and GRAL (ANR-10-LABX-49-01).

## Author contributions

W.M.S. planned, performed and analysed all biochemical and biophysical experiments. M.N. performed the selenomethionine phasing. M.R. performed the TMT labelling and MS analysis. F.S. analysed the MS data. S.C. conceived and directed the project and performed crystallographic analysis. W.M.S. and S.C. wrote the manuscript.

## Additional information

**Competing interests:** The authors declare no competing interests.

