## [Peer Review File · Nature Communications]

Reviewers' comments:

Reviewer #1 (Remarks to the Author):

In the manuscript by Shulze et al., the authors provide a structural analysis of human ARS2. ARS2 is a component of the nuclear cap complex that is critical for the processing, export, and degradation of many RNA pol II transcripts. A critical question in Pol II transcription is how the different nascent RNAs (mRNAs, miRNAs, snRNAs etc...) produced by Pol II recognized and targeted to the appropriate processing machinery and destination? Recently, ARS2 has emerged as a key adaptor molecule in the processing, export, and degradation of RNAPII RNAs. However, the mechanisms of how ARS2 achieves its multifunctional role are poorly understood. A partial crystal structure of the ARS2 orthologue, Serrate, has been determined and models of mammalian ARS2 have been generated based on this, but there are significant differences between the mammalian version of ARS2 and its plant counterpart- most importantly being that SERRATE lacks a RRM domain. Leaving us with at best a murky view of this critical player in transcription.

Shulze et al., now address this and in a structural tour-de- force provide the first look at the structure of human ARS2. The ARS2 protein has been refractory to crystallization as the protein has several inherent disordered regions interspersed between an N-terminal helix-turn-helix, a core helical bundle, an RRM domain, and a C-terminal leg, which resembles a Zn finger. To tackle this heterogeneity, the authors resort to limited proteolysis in which they digested the most of the unstructured pieces, leaving two folded fragments which they identified by mass spectrometry. They then co-expressed these fragments in E. coli to generate the first useable crystals. Further deletions of loop regions allowed further refinement to generate a reasonable first look at the ARS2 structure. The authors support the structural model they generate through mutation analysis coupled with in vitro interaction data to map the binding regions of a few ARS2 interactors, including the first demonstration that ARS2 binds directly to ssRNA through its RRM domain. They go on to study the interactome of ARS2 wt and several mutants in cell culture and show that a basic patch within the c-terminal leg of ARS2 is a platform for interacting with the NEXT, PAXT, and export complexes. The combination of biophysical, and biochemical data provide convincing support to the structural data that is presented. As with any good structural study, the work generates many more questions. At least now we have a structural framework on which to base future mechanistic exploration.

Overall, this is a very well described study that will be of interest to the RNA processing community. I do have a few comments that I would categorize as minor revisions that would improve the clarity of the study:

1) Figure 3c includes a delta 692-743 GSG mutant that is not described in the text or figure legend

2) Figure 4. The authors indicate in line 248 that " the binding site was further narrowed down to ARS2 147-270+494-763 (Figure 4a)..." However Figure 4a is labeled as ARS2 147-871 + FARB. It is important that the authors demonstrate that the fragments used to obtain the structure represent functional molecules. If this data is actually the two fragments, this could demonstrate functionality. However if it is not, it is important the authors include data to show that the two fragments used to make the structure are a functional version of the ARS2 protein by demonstrating the capacity to bind to ARS2 interactors.

3) Figure 5b. There are 8-10 proteins with a greater than 2 fold change and small p-value that are decreased when mutations are introduced within the ARS2 C-terminus. These should be labelled on the figure.

4) The authors identify a semi structured region which they label loop b and contains a EEE Helix. This appears to be a very flexible part of the protein that in some of the crystals is backed against

the basic region in the RRM domain. I think a discussion of the possible role of this region should be included in the discussion.

Reviewer #2 (Remarks to the Author):

Review Manuscript NSMB 155416_0

In the research manuscript entitled "Structural analysis of human ARS2, a platform for co-transcriptional RNA sorting" Wiebke Manuela Schulze, Frank Stein, Mandy Rettel, Max Nanao and Stephen Cusack present the crystal structure(s) of the human ARS2 core, which exhibits similarities to the plant homologue SERRATE, but also significant differences like an additional RRM domain.

Moreover, they present biochemical, biophysical and cell extract based pull-down data comparing wild type and structure-guided mutant forms of ARS2. The authors identify regions essential for direct interactions with cellular partners, like FLASH, NCBP3 and single-stranded RNA. Interestingly they show that FLASH and NCBP3 have overlapping binding sites in the C-terminal region of ARS2 and that CBC-ARS2-NCBP3 may form a ternary complex that is mutually exclusive with CBC-ARS PHAX. They thereby show that mutually exclusive higher order CBC-ARS2 complexes play a critical in determining Pol II transcript fate.

Overall this is a detailed work to understand the modes of interaction of the different binding partners to ARS2, which warrants publication in Nature Communication once the major points below have been addressed.

Major Points:

L129ff: A more elaborate definition and description of "a walking man" in fig.1 would make understanding easier (e.g. the "foot" used later on)

L144ff: "3AX1 is rather incomplete"; as the two structures are compared and, in my opinion, both are lacking quite some residues this should be mentioned in the text and/or an additional (supplemental) figure with the resolved and missing residues and would be very helpful to solidify this statement.

L741: Table 1

Major:

- the CC1/2 in the last shell for human ARS2 (171-270 and 408-763) as well as the I/sigma(I) deviate significantly from the "usual" values as they are pretty low. It would be helpful to support/justify the use of these data, e.g. by a paired refinement. In case this has been done it should be mentioned in the text (M&M?). Moreover, a plot of cc^* versus the other cc 's (work, free) would help to justify the use of the data in the last shell(s).

- as the average B-factors are high in two of the structures, the addition of the Wilson B-Factor would be helpful.

Minor:

- Cell dimensions: angles are missing the unit ($^{\circ}$), as well as this information is last column
- add the PDBid under "crystal"?

Figure 6:

Panel c and d are missing the correlation to the chromatogram and the column used (this is also not clearly stated in the materials and methods section as complex analysis is not mentioned).

Either the elution volume of the fractions need to be mentioned or the chromatogram should be shown (e. g. like in supplemental figure 1#)

Minor Points:

General:

- in the description of the amino acid residues there is an inconsistency sometimes using 3-letter code, sometimes 1-letter code.
- Some paragraphs miss the empty line at the end.

L25: cellular pull down? = Pulldown with cellular extracts?

L29: C-terminal leg: here nothing is mentioned about the overall structure, definition of this region come later.

L42: inconsistency of defining abbreviations used, sometimes yes, sometimes no (e.g. also L51: FADD and L62 PHAX; there are more).

L192/3: The description of this observation (loop only in one crystal and involved in contacts leads to the expectation of a consequence, so what is the significance of / conclusion from this observation?

L253: ARS2 gene has been cloned or the coding sequence?

L402: "with" too much. Delete?

L489/L546: g= grams, so maybe better xg

L499: L for liter

L628: "flash frozen": these words are causative for a general discussion among crystallographers. "frozen" indicates ice formation so usually "flash cooled" is the term used.

L673: "." missing

Responses to referees comments

Reviewer #1 (Remarks to the Author):

Overall, this is a very well described study that will be of interest to the RNA processing community. I do have a few comments that I would categorize as minor revisions that would improve the clarity of the study:

1) Figure 3c includes a delta 692-743 GSG mutant that is not described in the text or figure legend

This mutant is now correctly described in the text and legend. It is actually Δ 694-742GSG after correction of residue numbering according to the isoform of ARS2 we have used.

2) Figure 4. The authors indicate in line 248 that “ the binding site was further narrowed down to ARS2 147-270+494-763 (Figure 4a)...” However Figure 4a is labeled as ARS2 147-871 + FARB. It is important that the authors demonstrate that the fragments used to obtain the structure represent functional molecules. If this data is actually the two fragments, this could demonstrate functionality. However if it is not, it is important the authors include data to show that the two fragments used to make the structure are a functional version of the ARS2 protein by demonstrating the capacity to bind to ARS2 interactors.

There are four panels in Figure 4a. The third one along shows binding of FARB to ARS2 147-270+494-763. This demonstrates that neither the disordered insertion 271-407, nor the RRM 408-493 are required for FARB binding, consistent with it binding to the C-terminal leg/foot. We also have ITC results for the K_d of FARB binding to ARS2 171-270+408-763 as listed in Table 2 (but the ITC data itself is not shown).

However, we take the point that we do not explicitly highlight that the crystallized split constructs, ARS2 147-270+408-763 and/or 171-270+408-763, which lack the central unstructured loop, maintain the biological functionality of ARS2 147-871 or indeed 1-871. This is certainly true for ssRNA binding (Figure 3b) and FARB binding (Figure 4a). For NCBP3 binding we also have pull-downs showing that it binds equally well to ARS2 147-871, ARS2 171-270+408-763, ARS2 147-270+408-270 Δ loopA and ARS2 171-270+408-270 Δ loopA Δ loopB (now shown in new Supplementary Fig. 9), consistent with our observations that it binds to the C-terminal leg/foot.

We now include a statement in the discussion that the split constructs retain many of the known functions of the complete ARS2: ‘Overall, our biochemical studies demonstrate that the crystallised core ARS2^{147/171-270+408-763} has the same activity as ARS2¹⁴⁷⁻⁸⁷¹ with respect to several of the biological functions of ARS2 including interactions with ssRNA (Figure 3b), FLASH (Figure 4a, Table 2) and NCBP3 (Figure 6, Supplementary Fig. 9) and further deletions or mutations enable the interacting regions to be more precisely defined.’

3) Figure 5b. There are 8-10 proteins with a greater than 2 fold change and small p-value that are decreased when mutations are introduced within the ARS2 C-terminus. These should be labelled on the figure.

These are now labelled in Figure 5b

4) The authors identify a semi structured region which they label loop b and contains a EEE Helix. This appears to be a very flexible part of the protein that in some of the crystals is backed against the basic region in the RRM domain. I think a discussion of the possible role of this region should be included in the discussion.

We don't know the function of loop B and the EEEE helix, but we already made some comments in the results section on metazoan-specific loops and in the discussion, with respect to FARB. The former remark has now been moved to the first paragraph of the discussion: 'Concerning loop B, in most ARS2 crystal forms the negatively charged 'EEEE' helix binds onto the positively charged RRM β -sheet surface of a neighbouring molecule (Supplementary Fig. 4). Repetition of this inter-molecular contact generates a three dimensional network. However, it appears sterically impossible for one molecule to 'auto-inhibit' its own RNA binding surface in this way, although it could potentially happen inter-molecularly if ARS2 is at a sufficiently high concentration'. Concerning FARB, we say in the discussion 'However, these previous results (of O'Sullivan 2015) also implicated the β -sheet surface of the ARS2 RRM in FARB binding. It is intriguing that in most of our structures, the acidic 'EEEE helix' (566-EEVSAEEEEELLG-577) within loop B, which has a certain sequence similarity to FARB (931-DELEEGERSDSE-943) can make an inter-molecular interaction with the β -sheet surface of the RRM from another ARS2 molecule, suggesting that FARB might bind in the same place. Nevertheless, despite extensive co-crystallisation trials, including with the loop B/'EEEE' helix deletion mutant, we have never observed any electron density corresponding to FARB bound to ARS2 in a crystal, and all our biochemical evidence suggests that only the C-terminal foot is involved.'

To gain further insight it would be necessary to mutate or delete the loop in cells and attempt to discern a phenotype.

Reviewer #2 (Remarks to the Author):

In the research manuscript entitled "Structural analysis of human ARS2, a platform for co-transcriptional RNA sorting" Wiebke Manuela Schulze, Frank Stein, Mandy Rettel, Max Nanao and Stephen Cusack present the crystal structure(s) of the human ARS2 core, which exhibits similarities to the plant homologue SERRATE, but also significant differences like an additional RRM domain. Moreover, they present biochemical, biophysical and cell extract based pull-down data comparing wild type and structure-guided mutant forms of ARS2. The authors identify regions essential for direct interactions with cellular partners, like FLASH, NCBP3 and single-stranded RNA. Interestingly they show that FLASH and NCBP3 have overlapping binding sites in the C-terminal region of ARS2 and that CBC-ARS2-NCBP3 may form a ternary complex that is mutually exclusive with CBC-ARS PHAX. They thereby show that mutually exclusive higher order CBC-ARS2 complexes play a critical in determining Pol II transcript fate.

Overall this is a detailed work to understand the modes of interaction of the different binding partners to ARS2, which warrants publication in Nature Communication once the major points below have been addressed.

Major Points:

L129ff: A more elaborate definition and description of "a walking man" in fig.1 would make understanding easier (e.g. the "foot" used later on).

‘Walking man’ was the description used in the SERRATE structure paper, referring to the structure having a ‘body’ and ‘leading and lagging legs’. In response to the referee, we decided to leave out reference to the ‘walking man’ and simply refer to the N-terminal leg, C-terminal leg and C-terminal foot which are now all labelled in Figure 1a.

L144ff: "3AX1 is rather incomplete"; as the two structures are compared and, in my opinion, both are lacking quite some residues this should be mentioned in the text and/or an additional (supplemental) figure with the resolved and missing residues and would be very helpful to solidify this statement.

Our statement was perhaps misleading, but we meant it to refer to the fraction of crystallised construct that is actually visualized in the structure. The actual figures are: for human ARS2¹⁴⁹⁻²⁶⁰⁺⁴⁰⁸⁻⁷⁶³ composite model, 95.6 % visualized (457 out of 478, 21 missing residues in loop B) and for SERRATE (PDB:3AX1), 86 % visualized (300 out 350), missing residue in three different regions). Of course, there are other regions that were not included in either construct. For human ARS2 we actually only see 52 % of the complete protein (871 residues), for SERRATE it is 42% of the complete protein (720 residues). In the light of this, we now state (bottom of page 6): ‘It should be borne in mind that the observed crystal structures of human ARS2 and plant SERRATE represent 95.6 and 86 % of the crystallised constructs, respectively but only 52% (457/871) and 42% (300/720) of the complete proteins. In particular, for human ARS2, the construct lacks the N- and C-terminal extensions and large central region, all presumed to be unstructured (Figure 1b).’

L741: Table 1

Major:

- the CC1/2 in the last shell for human ARS2 (171-270 and 408-763) as well as the I/sigma(I) deviate significantly from the "usual" values as they are pretty low. It would be helpful to support/justify the use of these data, e.g. by a paired refinement. In case this has been done it should be mentioned in the text (M&M?). Moreover, a plot of cc versus the other cc's (work, free) would help to justify the use of the data in the last shell(s).*

We agree that the CC1/2 and I/σI for the P6₅ form of human ARS2 (171-270 and 408-763) are at the limit of normally accepted values. We have therefore decided to use the default criteria in AIMLESS which suggests cutting resolution when CC1/2 fall below 0.3. This raises the resolution to 3.54 Å where I/σI=1.01. We have taken the opportunity to improve the refinement statistics of this and the ΔloopB structures, since we found (following runs with PDB REDO) that introducing TLS parameters (1 per chain) lowers both R-factors considerably (see revised Table 1), improves the maps a bit, but does not change the structures significantly.

- as the average B-factors are high in two of the structures, the addition of the Wilson B-Factor would be helpful.

The Wilson B-Factor, as given by the PDB Validation server, has been added in Table 1. However in all our structures the Wilson B-factor is lower than the refined value, the more so the lower the resolution of the data. This we ascribe to the difficulty of correctly estimating the Wilson B-factor from low resolution data. We note that the PDB REDO mentioned above yielded the same average B-factor as we originally found.

Minor:

- Cell dimensions: angles are missing the unit ($^{\circ}$), as well as this information is last column
- add the PDB id under "crystal"?

Both done.

Figure 6:

Panel c and d are missing the correlation to the chromatogram and the column used (this is also not clearly stated in the materials and methods section as complex analysis is not mentioned). Either the elution volume of the fractions need to be mentioned or the chromatogram should be shown (e. g. like in supplemental figure 1#)

The elution volumes have been added to Fig. 6c and d. How complexes were analysed has been added to the Methods.

Minor Points:

General:

- in the description of the amino acid residues there is an inconsistency sometimes using 3-letter code, sometimes 1-letter code.

We now uniformly use the 1-letter code in text and figures.

- Some paragraphs miss the empty line at the end.

Corrected

L25: cellular pull down? = Pulldown with cellular extracts?

We have replaced this whole phrase 'We present biochemical, biophysical and cellular interactome data'

L29: C-terminal leg: here nothing is mentioned about the overall structure, definition of this region come later.

We replace this phrase simply by 'We show that FLASH and NCBP3 have overlapping binding sites on ARS2.... In any case the abstract was too long and has now been reduced to 150 words.

L42: inconsistency of defining abbreviations used, sometimes yes, sometimes no (e.g. also L51: FADD and L62 PHAX; there are more).

SRRT, FADD, PHAX and NCBP3 have now been defined in full.

L192/3: The description of this observation (loop only in one crystal and involved in contacts leads to the expectation of a consequence, so what is the significance of / conclusion from this observation?

In these lines we simply make observations about loop A. We now add ‘Interestingly T543 is a major phosphorylation site of human ARS2 (see discussion).’ In the discussion, we now say: ‘The exact function of loops A and B remain to be determined. However, interestingly T543 within loop A is reported to be a major phosphorylation site of hARS2 (<http://www.phosphosite.org/>) possibly by a proline directed kinase such as CK2 {Franchin, 2015 #297}. This may be a means of regulating ARS2 activity.’

That is as far as our knowledge goes at the moment. To gain further insight it would be necessary to mutate or delete the loop in cells and attempt to discern a phenotype.

L253: ARS2 gene has been cloned or the coding sequence?

The coding sequence. Corrected.

L402: "with" too much. Delete?

We think the ‘with’ should be there, as it would be if there were no ‘including’. ‘Nevertheless, despite extensive co-crystallisation trials (including) with the loop B deletion mutant...’.

L489/L546: g= grams, so maybe better xg

Done.

L499: L for liter

Done

L628: "flash frozen": these words are causative for a general discussion among crystallographers. "frozen" indicates ice formation so usually "flash cooled" is the term used.

This is very bizarre comment. If the sample is not frozen, then in what state is it in? There is indeed ice formation but it is amorphous ice.

L673: "." missing

Corrected

REVIEWERS' COMMENTS:

Reviewer #1 (Remarks to the Author):

I'm satisfied that the authors have adequately revised the manuscript to address concerns, however, they indicate they have modified figure 5b to include the labels of the proteins that showed >2-fold change but the figure I was provided looks the same as previous. Is this a mistake?

Reviewer #2 (Remarks to the Author):

In the revised version all major issues have been addressed. The manuscript is now even more suitable for publication.

Response to referee's comments:

As pointed out by the reviewer #1, we uploaded the old version of Figure 5b by mistake last time; this has now been rectified.